# Guiding Large Language Models via Directional Stimulus Prompting

**Zekun Li**[1]*, **Baolin Peng**[2], **Pengcheng He**[2], **Michel Galley**[2], **Jianfeng Gao**[2]†, **Xifeng Yan**[1]†

University of California, Santa Barbara[1]
Microsoft[2]
{zekunli, xyan}@cs.ucsb.edu
{bapeng,penhe,mgalley,jfgao}@microsoft.com

## Abstract

We introduce *Directional Stimulus Prompting*, a novel framework for guiding black-box large language models (LLMs) towards specific desired outputs. Instead of directly adjusting LLMs, our method employs a small tunable policy model (e.g., T5) to generate an auxiliary *directional stimulus prompt* for each input instance. These directional stimulus prompts act as nuanced, instance-specific hints and clues to guide LLMs in generating desired outcomes, such as including specific keywords in the generated summary. Our approach sidesteps the challenges of direct LLM tuning by optimizing the policy model to explore directional stimulus prompts that align LLMs with desired behaviors. The policy model can be optimized through 1) supervised fine-tuning using labeled data and 2) reinforcement learning from offline or online rewards based on the LLM's output. We evaluate our method across various tasks, including summarization, dialogue response generation, and chain-of-thought reasoning. Our experiments indicate a consistent improvement in the performance of LLMs such as ChatGPT, Codex, and InstructGPT on these supervised tasks with minimal labeled data. Remarkably, by utilizing merely 80 dialogues from the MultiWOZ dataset, our approach boosts ChatGPT's performance by a relative 41.4%, achieving or exceeding the performance of some fully supervised state-of-the-art models. Moreover, the instance-specific chain-of-thought prompt generated through our method enhances InstructGPT's reasoning accuracy, outperforming both generalized human-crafted prompts and those generated through automatic prompt engineering. The code and data are publicly available.[3]

## 1 Introduction

In recent years, a new paradigm has emerged in natural language processing (NLP) with the rise of large language models (LLMs) such as InstructGPT, ChatGPT [46], GPT-4 [45], PaLM [10], and others. These models exhibit emergent abilities [68] such as strong in-context learning and few-shot prompting capabilities, which were not present in previous "smaller" language models (LMs) like BERT [14], RoBERTa [37], GPT-2 [52], and T5 [53]. This shift in paradigm has led to remarkable advancements in NLP, with LLMs demonstrating impressive general-purpose power. However, due to commercial considerations and the risk of misuse, most LLMs do not publicly release their parameters and only allow users to access them through black-box APIs. While there also exist open-sourced LLMs, fine-tuning them for specific tasks or use cases can be computationally inefficient. In this scenario, the standard approach for utilizing LLMs to perform diverse tasks is crafting generalized

---

*Part of the work was done when Zekun Li was interning at Microsoft Research.
†Co-advise on this work.
[3]https://github.com/Leezekun/Directional-Stimulus-Prompting

37th Conference on Neural Information Processing Systems (NeurIPS 2023).

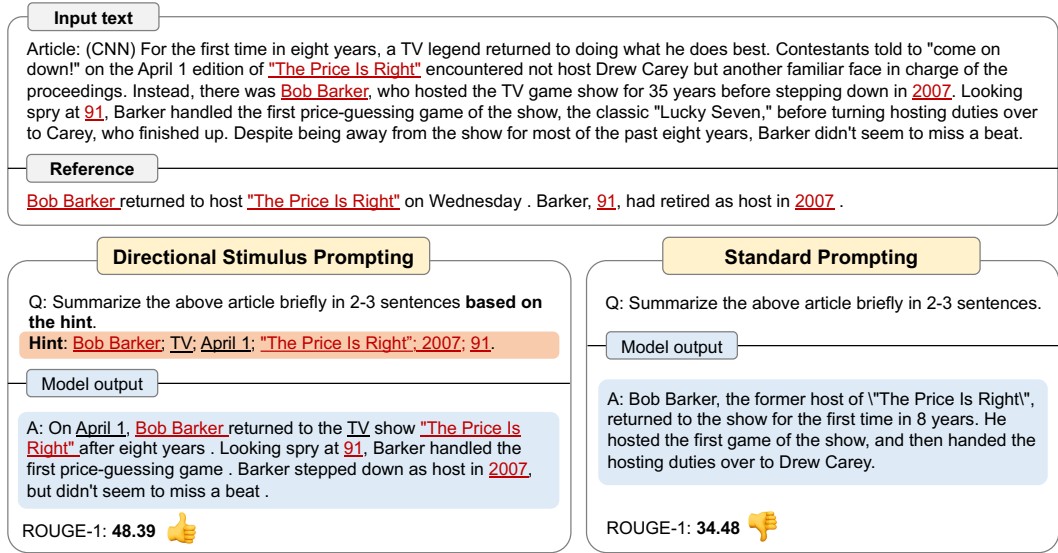

Figure 1: Comparison of our Directional Stimulus Prompting and the standard prompting method using LLMs such as ChatGPT for the summarization task. DSP utilizes directional stimulus/hints (highlighted in orange), which are keywords in this case, to provide instance-specific guidance to LLMs in generating summaries (highlighted in blue) that better align with the desired reference summary with higher ROUGE scores or other measures like human preferences.

task-specific prompts to query LLMs. While LLMs have demonstrated considerable performance on a wide range of language tasks, they still struggle to generate outputs that fully align with desired behaviors and directions on some specific tasks and use cases [16, 4].

Optimizing Large Language Models (LLMs) directly for specific tasks can be infeasible or inefficient for many users and developers, leading researchers to shift their focus towards prompt engineering and optimization. Prompt engineering approaches, which involve manually or automatically designing optimal task-specific natural language instructions and selecting appropriate training samples for demonstration in the prompt, have consequently gained the attention of researchers [6, 55, 79, 39]. Despite these efforts, the majority are centered on devising task-specific prompts, often falling short in steering LLMs to generate desired results on a per-instance basis.

To address the challenge, we propose a novel framework called **Directional Stimulus Prompting (DSP)**. This framework introduces a new component called the ***directional stimulus*** into the prompt to provide nuanced, instance-specific guidance and control over LLMs. Specifically, the directional stimulus prompt acts as ***hints*** and ***clues*** for the input query to guide LLMs toward the desired output. Notably, this differs from the methods that augment LLMs with additional knowledge retrieved from external sources [25, 60], as the directional stimulus prompt is generated solely based on the input query in our framework. Figure 1 compares our proposed prompting approach, DSP, with standard prompting for the summarization task. Our approach incorporates keywords in the prompt as the directional stimulus prompt to hint at key points the desired summary should cover. By providing this instance-specific guidance through directional stimulus prompts, LLMs can generate outputs that more closely align with the desired reference summary.

We utilize a relatively small and tunable LM (e.g., T5) as the policy model to generate the directional stimulus prompt for each input query. This approach enables us to sidestep the direct optimization of black-box LLMs by optimizing the small tunable policy model instead. We train the policy model through supervised fine-tuning (SFT) using a few collected labeled data. After supervised fine-tuning, we further optimize the policy model to explore better directional stimulus prompts with reinforcement learning (RL). During RL training, we aim to maximize the reward defined as downstream performance measures or any other measures of the LLM's output conditioned on the stimulus generated by the policy model.

Figure 2 provides the overview of our framework, using the summarization task as an illustrative example. We employ a compact, tunable policy model to generate the directional stimulus prompt,

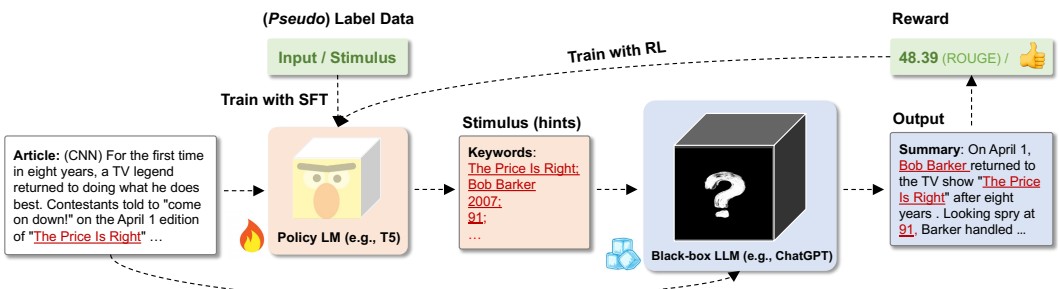

Figure 2: Overview of our proposed framework DSP, where we learn a small tunable policy model to generate the directional stimulus (keywords in this case) that provides input-specific guidance for the LLM toward the desired target. The policy model can be trained with SFT and/or RL, where the reward is defined as the downstream task performance measure, such as the ROUGE score for the summarization task, or other alignment measures like human preferences.

which specifies keywords that should be included in the LLM-generated summaries. The policy model can be trained with SFT and RL, where the reward is typically defined as the downstream task performance measure, such as the ROUGE score for the summarization task, or other alignment measures like human preferences.

Our framework can be flexibly adapted to a wide range of LMs and tasks by choosing the appropriate directional stimulus prompt, i.e., hints. We conducted experiments on summarization, dialogue response generation, and chain-of-thought reasoning tasks to evaluate the effectiveness of our framework. The results demonstrate that our DSP approach can effectively guide LLMs toward the desired targets with a small collection of labeled data. Specifically, we conduct experiments with the black-box LLMs: ChatGPT, Codex, and InstructGPT. For the policy model, we employ a 750M Flan-T5-Large [53, 11] and 220M T5-Base. For the summarization task, we use keywords as the directional stimulus, which hints at key points that the desired summary should include. Despite ChatGPT's already considerable performance, the policy model trained with only 4,000 samples from the CNN/Daily Mail dataset [43] improved the ROUGE and BLEU scores by 4-13%. For the dialogue response generation task, we train the policy model to generate dialogue acts that indicate the underlying intentions behind target responses on dialogues from the MultiWOZ dataset [7]. Guided by the policy model trained with only 80 dialogues, ChatGPT's performance improved by up to 41.4% in combined scores, achieving comparable or even better performance than some state-of-the-art models trained on the full dataset with 8,438 dialogues. For the chain-of-thought reasoning, we train the policy model to generate a trigger prompt for each input query to steer the LLM chain-of-thought reasoning, achieving better performance than the generalized hand-crafted prompts and those produced through the automatic prompt engineering approach [79], suggesting the effectiveness of our approach for automatic prompt engineering and optimization.

## 2   Directional stimulus prompting

For a downstream task, there is an input space $X$, a data distribution $\mathcal{D}$ over $X$, and an output space $Y$. Due to the strong in-context learning and few-shot prompting abilities, LLMs can perform diverse tasks and generate the output $y$ by including instructions that describe the task, a few demonstration examples, and the input query $x$ in the prompt [6]. However, such prompts cannot always steer LLMs toward desired outputs, especially when it comes to fine-grained instance-specific desired behaviors. For instance, in the case of the summarization task, the input $x$ is an article, and the output $y$ is the corresponding summary. Different summarizers have distinct styles and emphasize different aspects of an article [16]. In this case, it may not be enough to effectively steer LLMs toward generating summaries that closely match reference summaries relying solely on task-specific instructions or demonstration examples to describe such nuanced differences for each sample.

To this end, our Directional Stimulus Prompting (DSP) approach introduces a small piece of discrete tokens $z$ named "*directional stimulus*" into the prompt, which acts as hints and clues to provide LLMs with fine-grained guidance toward the desired direction. For example, for the summarization task, the directional stimulus $z$ might consist of keywords that should be included in the desired

summary. To generate this stimulus for each input query, we use a small tunable policy language model, $p_{\text{POL}}(z|x)$. We then use this generated stimulus, $z$, along with the original input, $x$, to construct the prompt that steers the LLM toward generating its output, $p_{\text{LLM}}(y|x, z)$. It's important to note that the parameters of the LLM, $p_{\text{LLM}}$, are kept frozen, as they are either inaccessible or inefficient to tune. Overall, when using the LLM with DSP to perform a downstream task, the output is obtained via $y \sim p_{\text{LLM}}(\cdot|x, z), z \sim p_{\text{POL}}(\cdot|x)$.

## 2.1 Supervised fine-tuning

To train the policy model that generates directional stimulus for LLMs, we first perform supervised fine-tuning (SFT) on a pre-trained LM (e.g., T5, GPT-2, etc) on a small collection of labeled data. To collect the data, we could heuristically select or annotate the "pseudo-stimulus" $z^*$ for each input query $x$ and target output $y$ pair based on the downstream task. For example, for the summarization task, we use keywords that the reference summary includes as pseudo-stimulus, while for the dialogue response generation task, we use dialogue acts that indicate the underlying meaning of the desired system response (see Section 3 for details). The resulting dataset $\mathcal{D}' = \{(x, z^*)\}$ consists of input-stimulus pairs. We then fine-tune the policy model by maximizing the log-likelihood:

$$\mathcal{L}_{\text{SFT}} = -\mathbb{E}_{(x,z^*)\sim\mathcal{D}'}\log p_{\text{POL}}(z^*|x). \tag{1}$$

Supervised fine-tuning can provide a good initial point for the policy model. However, it is important to note that the heuristically selected or annotated pseudo-stimulus may not always be optimal, and the supervised fine-tuned policy model may not generate the most preferred directional stimulus for the LLMs toward the desired outputs. To overcome this limitation, we can also incorporate reinforcement learning (RL) to further fine-tune the policy model. By directly optimizing the LLM's output toward desired targets, RL training enables the policy model to explore and generate more effective directional stimulus.

## 2.2 Reinforcement learning

**Optimization objective** Our goal is to steer the LLM's generation toward the desired target by maximizing an alignment measure $\mathcal{R}$, which can take various forms such as downstream task performance measures (e.g., ROUGE score for summarization), human preferences, or other customized measures. Mathematically, we aim to maximize the below objective:

$$\mathbb{E}_{x\sim\mathcal{D},z\sim p_{\text{POL}}(\cdot|x),y\sim p_{\text{LLM}}(\cdot|x,z)}[\mathcal{R}(x, y)]. \tag{2}$$

Since the parameters of the black-box LLM are not accessible or tunable, we resort to optimizing the policy model to generate the directional stimulus that guides the LLMs' generation toward maximizing the objective. To achieve that, we define another measure $\mathcal{R}_{\text{LLM}}$ that captures how well the LLM performs when conditioned on a given stimulus $z$:

$$\mathcal{R}_{\text{LLM}}(x, z) = \mathcal{R}(x, y), y \sim p_{\text{LLM}}(\cdot|x, z). \tag{3}$$

This allows us to cast the original objective of maximizing $\mathcal{R}$ into optimizing the policy model to generate stimulus that maximizes $\mathcal{R}_{\text{LLM}}$. By doing so, the LLM is effectively used as an evaluation function to guide the policy model toward generating more effective directional stimulus. Thus, the optimization objective for LLMs in Equation 2 is equal to the optimization objective for the policy model:

$$\max_{p_{\text{POL}}}\mathbb{E}_{x\sim\mathcal{D},z\sim p_{\text{POL}}(\cdot|x)}[\mathcal{R}_{\text{LLM}}(x, z)]. \tag{4}$$

**RL formulation** However, the above optimization is intractable for the policy model. To address the issue, we formulate the policy model optimization as an RL problem and employ proximal policy optimization (PPO) [59]. We use the policy model to initialize a policy network $\pi_0 = p_{\text{POL}}$ and then update $\pi$ using PPO. The process that the policy model generates a sequence of tokens as stimulus $z$ can be seen as a Markov decision process (MDP) $\langle \mathcal{S}, \mathcal{A}, r, \mathcal{P}\rangle$ with a state space $\mathcal{S}$, action space $\mathcal{A}$, reward function $r$, and state-transition probability $\mathcal{P}$. In each time step $t$ of an episode, the agent selects an action (token) from the vocabulary $\mathcal{V}$ according to the distribution of the current policy network $\pi(z|x, z_{<t})$. The episode ends when an end-of-sequence token is selected, and the stimulus $z$ is generated. We can fine-tune the policy network $\pi$ by optimizing the reward $r$:

$$\mathbb{E}_\pi[r] = \mathbb{E}_{x\sim\mathcal{D},z\sim\pi(\cdot|x)}[r(x, z)]. \tag{5}$$

**Reward function** Recall that our goal is to maximize the objective in Equation 4, which can be used as the reward $r$. To keep the policy network $\pi$ from moving too far from the initial policy model $p_{\text{POL}}$, we also add a KL-divergence penalty reward. Therefore, the final reward becomes:

$$r(\boldsymbol{x}, \boldsymbol{z}) = \mathcal{R}_{\text{LLM}}(\boldsymbol{x}, \boldsymbol{z}) - \beta \log \frac{\pi(\boldsymbol{z}|\boldsymbol{x})}{p_{\text{POL}}(\boldsymbol{z}|\boldsymbol{x})}. \tag{6}$$

Following [80, 54], we dynamically adapt the coefficient $\beta$ during training:

$$\boldsymbol{e}_t = \text{clip}\left(\frac{\text{KL}(\pi_t, p_{\text{POL}}) - \text{KL}_{\text{target}}}{\text{KL}_{\text{target}}}, -0.2, 0.2\right), \tag{7}$$

$$\beta_{t+1} = \beta_t \left(1 + K_\beta \boldsymbol{e}_t\right). \tag{8}$$

**Implementation** To optimize the policy network $\pi$, we use the NLPO version of PPO from [54], which is specifically designed for language generators. To address the issue of large action spaces in PPO, NLPO learns to mask out less relevant tokens in the vocabulary using top-$p$ sampling. This technique restricts the action space to the smallest set of tokens whose cumulative probability is greater than the given probability parameter $p$, which we set to 0.9 in our experiments. Both the policy network $\pi$ and value network are initialized from the supervised fine-tuned policy model $p_{\text{POL}}$, with the final layer of the value network randomly initialized to output a scalar value using a regression head.

## 3 Experiments

Our proposed framework DSP can be flexibly applied to various types of LMs and generation tasks. In this work, we focus on 1) summarization, 2) dialogue response generation, and 3) chain-of-thought reasoning tasks. We mainly use pre-trained T5 or Flan-T5 [53, 11] to initialize the policy model and experiment with the black-box LLMs including **ChatGPT (`gpt-3.5-turbo`)**, **Codex (`code-davinci-002`)**, and **InstructGPT (`text-davinci-002`)**.

### 3.1 Summarization

Recent studies [16, 75, 4] have shown that LLMs, such as GPT-3, InstructGPT, and ChatGPT, are capable of generating high-quality summaries with zero- or few-shot prompting. However, their reference-based evaluation benchmark performances, such as ROUGE scores, still lag behind fine-tuned methods, indicating that the generated summaries may not completely match the style and emphasis of the reference summaries. In our experiments, we seek to guide LLMs to generate summaries that more closely align with the reference summaries by providing keywords that should be mentioned in the desired summaries as hints. We evaluate the effectiveness using metrics that compare the generated summaries against reference summaries. Notably, other desired directions, such as better alignment with human preferences, can also be pursued.

**Dataset and evaluation** We conduct our experiments on the CNN/Daily Mail dataset, a widely-used news summarization benchmark. To keep the cost of API usage low, we train on a subset of 1,000, 2,000, and 4,000 article-summary pairs from the total 287,113 samples in the training set. For evaluation, we randomly select 500 samples, following previous work [16, 65], which has been proven to provide sufficient statistical power [8]. We use the overlap-based metrics, including ROUGE [33], BLEU [47], and Meteor [3], and the similarity-based metric, BERTScore [74], to compare the generated summaries with the references. The reported evaluation scores are averaged over three inferences of ChatGPT for each query, using a temperature of 0.7 and top_p of 1.0. We use the same three demonstration examples in the prompt for standard prompting and add keywords as directional stimulus in the prompt for our approach, DSP. The exact prompts used in our experiments are provided in the Appendix.

**Supervised fine-tuning details** We use keywords as the pseudo-stimulus to train the policy model with supervised fine-tuning as discussed in Section 2.1. To collect the data, we employ textrank [41, 5] to automatically extract the keywords from the article and summary and only keep those that appear in the reference summary. As a result, we obtain a list of extracted keywords for each article-summary pair in the dataset. To convert them into a sentence that serves as the stimulus, we concatenate them using a split token ";", resulting in the stimulus formated as "*[Keyword1]; [Keyword2]; ... ;*

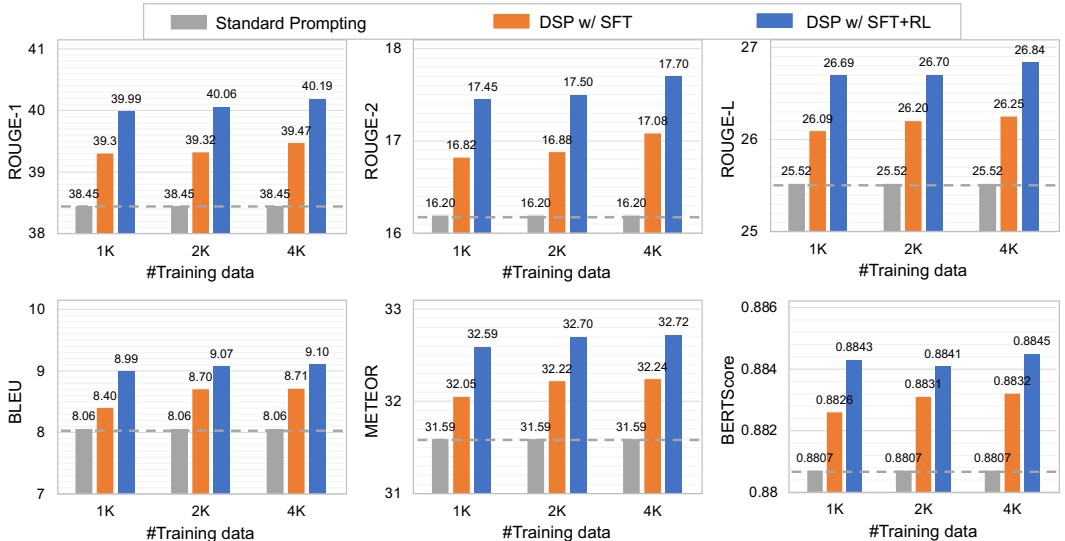

Figure 3: Performance comparison of ChatGPT with standard prompting and DSP trained with SFT and SFT+RL, using varying numbers of training samples from the CNN/Daily Mail dataset.

*[KeywordN]*.". We use the constructed article-stimulus pairs to train the policy model via supervised fine-tuning. The input format for training is "*Extract the keywords: [Article]*", while the output is the target stimulus consisting of keywords. The policy model was trained for 5 epochs with a $2 \times 10^{-5}$ learning rate.

**RL training details** As we aim to guide ChatGPT in generating summaries that more closely match the reference summaries, we adopt the automatic reference-based metric scores as the alignment measure reward. Specifically, we calculate the ROUGE-Avg score between the generated summaries and the reference summaries as the reward, with a rescaling coefficient of 10. We experimentally found that other automatic evaluation metrics, such as BLEU and Meteor, perform similarly. To reduce variance, we generate four outputs per input query using ChatGPT with a temperature of 0.7 and compute the average reward. Additionally, we assign a step-wise reward, which we found could improve the efficiency and stability of the training process. Specifically, the policy model generates a sequence of keywords in each episode, during which we assign a reward of 1 if a keyword appears in the reference summary and a penalty reward of -0.2 is given otherwise. We train the policy network for 51k episodes, with 5 epochs per batch, a batch size of 8, and a learning rate of $2 \times 10^{-6}$. The $KL_{target}$ and $\beta_0$ in Equation 7 are set to 0.5 and 0.005, respectively.

**Results** We evaluate the performance of ChatGPT with standard prompting and our approach DSP trained with SFT or SFT and then RL (SFT+RL) on varying sizes of training data and present the results in Figure 3. As can be seen, all the evaluation scores improve with our proposed DSP compared with standard prompting. Specifically, the supervised fine-tuned policy model generates the stimulus that effectively guides ChatGPT to generate summaries that closely align with the reference summaries, leading to improved benchmark performance. Furthermore, the additional fine-tuning of the policy model with RL results in further performance improvement, indicating the effectiveness of RL in exploring better directional stimulus that maximizes the reward. As the size of the training data increases, the performance improvement becomes more significant. Despite using a small collection of only 1,000 to 4,000 samples to keep API usage costs low, our DSP approach still

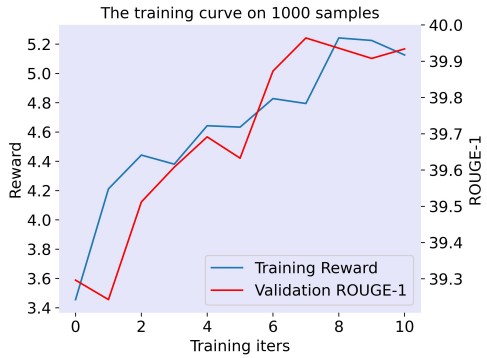

Figure 4: Training curve on 1000 samples from the CNN/Daily Mail dataset.

consistently enhances ChatGPT's ROUGE, BLEU, and Meteor scores by 1-2 in absolute points, even though ChatGPT has already achieved considerable performance. However, due to the discrepancy between the semantic-based metric BERTScore and the overlap-based metric ROUGE, which are used as the reward, the improvement in BERTScore after RL training may be relatively less significant. Figure 4 presents the change of training rewards and ROUGE-1 score on the validation set during the training process on 1,000 samples. We can see that the performance is closely related to the training rewards, and the training is relatively stable using the NLPO algorithm.

## 3.2 Dialogue response generation

In recent years, there has been a rise in LLM-based chatbots such as ChatGPT[4] and Sparrow [5]. These chatbots are typically targeted at open-domain conversations to engage with users on a wide range of topics without a specific goal in mind. However, these chatbots still face challenges in handling task-oriented dialogues where they need to assist users in completing specific goals or tasks, such as making reservations or ordering food [4, 22]. Unlike open-domain conversations, task-oriented dialogues often require the chatbot to follow task-specific business logic and respond based on reliable information from API calls or database queries. To address this limitation, we train a small policy model to learn the underlying dialogue policy from the training data and thus guide the LLMs in generating reliable system responses that assist users in completing tasks.

**Dataset and evaluation** We conduct experiments on the popular task-oriented dialogue dataset MultiWOZ [7], including both the MultiWOZ2.0 (the original version) and MultiWOZ2.1 version [15]. The dataset provides annotations for user utterances, dialogue acts, and system responses for each dialogue turn. The goal is to generate the system response given the history dialogue context as input. We utilize the dialogue act, which represents the communicative intention of the target system response, as the pseudo-stimulus for our experiment. There are 8,438 dialogues in the training set. We only use 1% (80 dialogues) and 10% (800 dialogues) to train the policy model and evaluate the performance on the full validation and test set, which contains 1,000 dialogues. We use the standard evaluation metrics: **Inform**, which measures the rate that the appropriate entity that satisfies the user's requirements is provided; **Success**, which measures the rate that all requested attributes are answered; **BLEU**: the corpus-level BLEU score with reference responses; and an overall measure **Combined score** = (Inform+Success)×0.5+BLEU. Likewise, we report the average score over three inferences. We use the same three demonstration examples when using DSP or standard prompting.

**Supervised fine-tuning details** To conduct supervised fine-tuning on the policy model, we format the input of each sample as *Translate dialogue to dialogue action: [Dialogue context]*", with the target being the verbalized dialogue acts in the same format as [77, 63]. For instance, a dialogue act *<hotel, inform, choice>, <hotel, inform, type>, <hotel, request, area>* will be converted to "*[hotel] [inform] choice type [request] area*", which indicates that the system should inform available hotel choices and their types and ask for the area that the user would like (see the Appendix for examples). Note that the provided dialogue act annotations may not be the only valid dialogue act for the same dialogue content [77], and thus we hope to explore diverse valid dialogue acts (directional stimulus) through RL training.

**RL training details** The evaluation metrics Success and Inform rates are defined at the dialogue level, while the BLEU score is computed on the corpus level. However, our training and inference on conducted on the turn level. We thus use the sentence-level SacreBLEU [51] score as the reward. Same as in the summarization experiments, we generate four outputs per input using the LLM with a temperature of 0.7. The policy network is trained 52k episodes, 5 epochs per batch with a batch size of 8 and a learning rate of $2 \times 10^{-6}$. Since the generated dialogue acts should adhere to the business logic and ontology, we ensure that the updated policy network does not deviate significantly from the original policy model. We thus set the $KL_{target}$ and $\beta_0$ in Equation 7 as 0.2 and 0.01, respectively. During training, we use top-$k$ sampling and set $k$ to 50 to explore the action space. During inference, we use beam search decoding with a beam size of 5.

**Results** We evaluate the impact of our approach DSP on Codex and ChatGPT and compare the performance with several representative task-oriented dialogue models trained on the full training set (8438 dialogues), including DAMD [77], MinTL [34], Soloist [49], SimpleTOD [21], DoTS [23], PP-

---

[4]https://openAI.com/blog/chatgpt
[5]https://www.deepmind.com/blog/building-safer-dialogue-agents

Table 1: Response generation performance of different methods on the MultiWOZ 2.0&2.1 datasets, where Succ. and Comb. denote the Success and Combined Score metrics, respectively.

| Method | #Training data | MultiWOZ 2.0 | | | | MultiWOZ 2.1 | | | |
|---|---|---|---|---|---|---|---|---|---|
| | | Inform | Succ. | BLEU | Comb. | Inform | Succ. | BLEU | Comb. |
| *Codex* | | | | | | | | | |
| Standard Prompting | - | 76.7 | 41.5 | 7.7 | 66.8 | 74.2 | 41.9 | 7.8 | 65.9 |
| DSP w/ SFT | 1% (80) | 74.9 | 66.3 | 11.1 | 81.7 | 72.0 | 66.0 | 11.3 | 80.1 |
| DSP w/ SFT+RL | 1% (80) | 91.0 | 76.0 | 9.8 | 93.3 | 89.7 | 78.6 | 9.4 | 93.4 |
| DSP w/ SFT | 10% (800) | 79.4 | 71.9 | 11.3 | 87.0 | 72.0 | 67.0 | 13.1 | 82.6 |
| DSP w/ SFT+RL | 10% (800) | 96.0 | 86.9 | 10.7 | 102.2 | 94.0 | 86.0 | 9.2 | 99.2 |
| *ChatGPT* | | | | | | | | | |
| Standard Prompting | - | 71.8 | 44.1 | 10.5 | 68.4 | 72.8 | 44.2 | 10.4 | 68.9 |
| DSP w/ SFT | 1% (80) | 76.6 | 66.5 | 11.2 | 82.8 | 76.0 | 64.3 | 11.3 | 81.4 |
| DSP w/ SFT+RL | 1% (80) | 90.9 | 82.2 | 10.2 | 96.7 | 87.3 | 78.7 | 10.7 | 93.7 |
| DSP w/ SFT | 10% (800) | 72.7 | 64.7 | 11.8 | 80.5 | 75.0 | 67.7 | 12.6 | 83.9 |
| DSP w/ SFT+RL | 10% (800) | 95.3 | 82.3 | 10.9 | 99.6 | 95.0 | 84.0 | 10.7 | 100.2 |
| *Fully supervised TOD models* | | | | | | | | | |
| DAMD [77] | 100% (8438) | 76.3 | 60.4 | 16.6 | 85.0 | - | - | - | - |
| MinTL [34] | 100% (8438) | 84.9 | 74.9 | 17.9 | 97.8 | - | - | - | - |
| Soloist [49] | 100% (8438) | 85.5 | 72.9 | 16.5 | 95.7 | - | - | - | - |
| SimpleTOD [21] | 100% (8438) | 84.4 | 70.1 | 15.0 | 92.3 | 85.0 | 70.5 | 15.2 | 93.0 |
| DoTS [23] | 100% (8438) | 86.6 | 74.1 | 15.1 | 95.5 | 86.7 | 74.2 | 15.9 | 96.3 |
| PPTOD [63] | 100% (8438) | 89.2 | 79.4 | 18.6 | 102.9 | 87.1 | 79.1 | 19.2 | 102.3 |
| UBAR [72] | 100% (8438) | 95.4 | 80.7 | 17.0 | 105.1 | 95.7 | 81.8 | 16.5 | 105.3 |
| GALAXY [19] | 100% (8438) | 94.4 | 85.3 | 20.5 | 110.4 | 95.3 | 86.2 | 20.0 | 110.8 |

TOD [63], UBAR [72], and GALAXY [19]. Table 1 summarizes the overall performance comparison, from which we obtain the following observations: (1) Our approach DSP significantly improves the success and inform rates of Codex and ChatGPT, indicating that they better understand the scenario and generate appropriate responses that help users in completing their tasks. (2) However, there is no improvement in the corpus-level BLEU score, possibly because the LLMs generate responses with different speaking styles and vocabulary since they do not see oracle system responses. Nevertheless, the high success and inform rates demonstrate the usefulness of our approach in delivering helpful and reliable responses. (3) Increasing the number of supervised fine-tuning samples does not guarantee performance improvement, but further fine-tuning the policy model using RL consistently provides performance gains. This suggests that RL training encourages the policy model to explore more model-preferred stimulus, while supervised fine-tuning may merely generate stimulus closely aligned with the pseudo-labeled data, which is not necessarily optimal. (4) Our approach achieves notable success with only 80 dialogues, surpassing several fully trained TOD models, particularly in terms of Success and Inform rates. With 10% of the training data (800 dialogues), our approach delivers comparable performance to current SOTA methods trained with full training data (8438 dialogues). We have also provided the performance of these compared methods in the low-resource settings (1% and 10%) and a running example in the Appendix.

### 3.3 Chain-of-Thought reasoning

While current methods primarily utilize generalized task-specific prompts, LLMs exhibit sensitivity to these prompts. Existing studies [69, 26, 79] illustrate that the performance of LLMs can vary significantly based on the prompt used. Consequently, a substantial portion of earlier work has been dedicated to either manually [56] or automatically [61, 79] crafting prompts. However, these studies largely concentrate on task-specific prompts, which may not be optimal for every instance of a task. In our experiment, we employ our approach to generate instance-specific prompts to elicit Chain-of-Thought (CoT) reasoning. Specifically, we train a policy model (t5-base) to generate instance-specific CoT trigger prompts, such as "*Let's think step by step*", to prompt varying samples.

**Dataset and evaluation** We adopted the experimental setup from previous work [26, 79], where we tested zero-shot CoT reasoning abilities of InstructGPT (text-davinci-002) with different trigger prompts. There are 600 examples in the MultiArith dataset [57], which we divided into

Table 2: Zero-shot chain of thoughts reasoning accuracy (%) of text-davinci-002 with different prompts. *Our approach trains a policy model to generate instance-specific prompt triggers, which are compared to the task-specific prompts in [26, 79].

| No. | Category | Chain-of-Thought Trigger Prompt | MultiArith | AQuA |
|-----|----------|--------------------------------|------------|------|
| 1 | Human-Designed | *Let's think step by step.* | 79.6 | 31.9 |
| 2 | | *We should think about this step by step.* | 81.2 | 28.7 |
| 3 | | *First,* | 78.0 | 38.2 |
| 4 | | *Before we dive into the answer,* | 54.8 | 27.2 |
| 5 | | *Proof followed by the answer.* | 58.4 | 37.8 |
| 6 | | *Let's think step by step in a realistic way.* | 59.6 | 33.9 |
| 7 | | *Let's think step by step using common sense and knowledge.* | 80.0 | 34.3 |
| 8 | | *Let's think like a detective step by step.* | 73.6 | 24.0 |
| 9 | | *Let's think about this logically.* | 75.2 | 34.7 |
| 10 | | *Let's think step by step. First,* | 78.8 | 32.3 |
| 11 | | *Let's think* | 56.8 | 38.2 |
| 12 | | *Let's solve this problem by splitting it into steps.* | 72.4 | 33.2 |
| 13 | | *The answer is after the proof.* | 42.8 | 34.3 |
| 14 | | *Let's be realistic and think step by step.* | 69.6 | 29.9 |
| 15 | APE [79] | *Let's work this out in a step by step way to be sure we have the right answer.* | 81.6 | 34.3 |
| 16 | DSP w/ SFT | (*Generated instance-specific prompt) | 75.2 | 35.8 |
| 17 | DSP w/ SFT+RL | (*Generated instance-specific prompt) | **84.0** | **38.6** |

300/50/250 for training/validation/test set. As for the AQuA dataset [35], we use the standard test set with 254 samples, 300 samples from the standard training set for our training, and 100 samples for the standard validation set for our validation. We report the reasoning accuracy.

**Supervised fine-tuning details** For supervised fine-tuning (SFT), we first run inference on the training set with the 14 human-crafted prompts tested in [26], respectively. We then selected those prompt and query pairs which resulted in a correct CoT reasoning outcome to form the training set for SFT. These query-prompt pairs were used to train a t5-base policy model for 2 epochs, with the model input being the query instance and the target output a trigger prompt.

**RL training details** After SFT, the prompts generated by the policy model were used to trigger InstructGPT for zero-shot CoT prompting. Reasoning accuracy was utilized as the reward for reinforcement learning (RL). A reward of 1 was assigned for correct reasoning results and 0 otherwise. We conducted 20 training iterations (106k episodes), with 5 epochs per batch, a batch size of 8, and a learning rate of 2e-6. The parameters for $KL_{target}$ and $\beta_0$ were set to 0.5 and 0.001, respectively.

**Results** We compare the performance of using our generated instance-specific prompts with using the 14 human-crafted prompts which we used as the pseudo-stimulus to constitute the training set for SFT and also the prompt automatically discovered by the APE approach [79]. Note that all these 15 prompts are generalized task-specific and are used for the whole test set while ours are instance-specific. The performance comparison is shown in the Table 8. As can be seen, InstructGPT's performance varies significantly when using different task-specific prompts. Compared to the 14 task-specific human-designed prompts, DSP enhances the performance with instance-specific prompts. It also outperforms the prompt discovered by the APE approach, suggesting the effectiveness of our approach for automatically prompt engineering and optimization. Solely relying on supervised fine-tuning of the policy model with the dataset comprising the 14 human-designed prompts doesn't lead to its peak performance. After fine-tuning with RL, the policy model is encouraged to explore better instance-specific trigger prompts, further improving performance.

## 4    Related work

**Black-box large language models** Recent years have witnessed the emergence of LLMs such as GPT-3 [6], Codex [9], InstructGPT, ChatGPT [46], PaLM [10], and LaMDA [66], which show significant promise in the field of NLP. These LLMs typically have a large number of parameters and require vast amounts of training data. Due to their scaling, these models have exhibited many emergent abilities, such as in-context learning, few-shot prompting, chain-of-thought prompting,

and instruction following [6, 46, 69]. However, most LLMs are not open-sourced and can only be accessed via black-box APIs, through which the users send prompt queries and receive responses. While there exist open-source LLMs such as OPT-175B [73] and Bloom [58], their local execution and fine-tuning require significant computational resources that may be infeasible for most researchers and users. However, despite their considerable performance on various tasks, LLMs often fall short of generating outputs that fully align with desired outputs on specific downstream tasks and use cases [16, 42, 18]. Our approach seeks to address this limitation by introducing directional stimulus generated by a small tunable LM into the prompt to provide more fine-grained guidance and control over black-box LLMs.

**Prompt optimization and engineering** Efficiently optimizing pre-trained LMs on downstream tasks by finding optimal prompts has been a focus of prior research. One approach involves tuning soft prompts, which are continuous embedding vectors that can be optimized using gradient descent methods [32, 30, 67, 2, 64]. However, the requirements of gradients and the challenge of passing gradients and continuous prompts through black-box APIs, making them less practical for the black-box LLMs. Researchers have also tried to seek optimal prompts by designing task-specific natural language instructions and selecting proper training samples as in-context demonstrations in the prompt. These methods include manual engineering [50, 6, 56], editing [61, 76], reinforcement learning [13, 39], and automatic generation [79]. Despite these efforts, such prompts are not always effective at steering LLMs to generate desired outputs, especially for fine-grained instance-specific behaviors that are difficult to describe using task-specific instructions and demonstration examples. To address this limitation, our approach is able to provide more **fine-grained instance-specific** guidance through the directional stimulus prompts (hints) generated by a small tunable policy model which could be optimized with supervised fine-tuning and reinforcement learning.

**Controllable text generation** The control of language models (LMs) has been extensively studied. Early approaches fine-tuned LMs on datasets containing desired attributes [17]. [24] proposed class-conditioned LMs, generating text with predefined control codes. However, direct LM training is costly. To address this, PPLM [12] trains an attribute model and passes gradients to control generation. GeDi [27] and DExperts [36] use class-conditional distributions as generative discriminators to guide generation, reducing computation complexity. These methods require either additional LM training or internal gradients and logistics, making them not applicable to black-box LLMs. Our approach proposes a solution to control black-box LLMs by inserting directional stimulus into the input query prompt and optimizing based on the return output.

**Reinforcement learning for NLP** Reinforcement learning has been successfully applied to various NLP tasks, such as syntactic parsing [44, 29], machine translation [71, 28], summarization [48, 62], conversational systems [31], etc. Language models define probability distributions over tokens in their vocabulary, and the text generation problem can be naturally formulated as selecting an action in an RL setting. Therefore, there have been extensive research efforts on optimizing LMs with RL, usually by aligning them with human preferences [80, 70, 40, 62]. For example, the LLM InstructGPT [46] is optimized with RL to better follow users' instructions and intent. In contrast with these works that directly update the LLMs to align with human preferences, our work optimizes a small policy model that generates text (stimulus) to guide LLMs to generate more human-preferred output instead of directly optimizing the LLMs, bypassing the inefficient LLM's optimization.

## 5 Conclusions and future work

In this paper, we introduce *Directional Stimulus Prompting* (DSP), a new prompting framework to provide black-box LLMs with fine-grained and instance-specific guidance toward the desired outputs. We use a tunable policy model to generate the directional stimulus to provide such guidance and convert the optimization of black-box LLMs to that of the policy model. Experimental results demonstrate the effectiveness of our approach in controlling and guiding black-box LLMs via automatic prompt engineering and optimization. Furthermore, the generated stimulus provides valuable insights and interpretations of LLMs' behaviors. In this work, we use heuristically selected or annotated pseudo-stimulus data for supervised fine-tuning of the policy model. For future work, we hope to explore the possibility of using a "machine language" between the policy model and the LLMs that might not be intuitively preferred by humans but can better convey guidance information, as well as other forms of directional stimulus beyond text.

## Acknowledgments and Disclosure of Funding

This research was partly sponsored by the DARPA PTG program (HR001122C0009). Any opinions, findings, conclusions, or recommendations expressed in this paper are those of the authors and do not necessarily reflect the views of funding agencies.

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

# A    Implementation Details

## A.1    Summarization

We use the representative benchmark dataset CNN/Daily Mail for news summarization [43]. This dataset contains 287,113 training examples, 13,368 validation examples, and 11,490 test examples. To keep the API usage cost low, we use a subset of 1,000, 2,000, and 4,000 for training, 500 for validation, and 500 for testing. Each example in the dataset consists of a news article along with its corresponding highlight/summary written by human authors. In order to train the policy model through supervised fine-tuning, we employed the textrank [41] algorithm to automatically extract keywords from each article and only retained those mentioned in the corresponding reference summary. We initialize the policy model using the 780M FLAN-T5-large model [11, 53], and use it to guide the black-box LLM ChatGPT. The hyperparameters used in our experiments are detailed in Table 3. All the experiments are run on a server equipped with 8 NVIDIA RTX A6000 GPUs.

| Model Params | Hyperparameter values |
|---|---|
| Supervised fine-tuning (SFT) | batch size: 8
epochs: 5
learning rate: 0.00002
learning rate scheduler: linear
weight decay: 0.01 |
| RL (NLPO) | steps per update: 5120
total number of steps: 51200
batch size: 8
epochs per update: 5
learning rate: 0.000002
entropy coefficient: 0.0
initial kl coeff: 0.005
target kl: 0.5
discount factor: 0.99
gae lambda: 0.95
clip ratio: 0.2
value function coeff: 0.5
rollouts top k: 100
top mask ratio: 0.9
target update iterations: 20 |
| Tokenizer | padding side: right
truncation side: right
max length: 512 |
| Policy model decoding | sampling: True
temperature: 0.7
min length: 10
max new tokens: 80 |
| LLM decoding | sampling: True
temperature: 0.7
top_p: 1.0
max new tokens: 180 |

Table 3: Hyperparameters for experiments on the CNN/Daily Mail dataset.

## A.2    Dialogue response generation

The MultiWOZ dataset is a widely-used task-oriented dialogue dataset consisting of 8,438 dialogues for training, 1,000 dialogues for validation, and 1,000 dialogues for testing. For each turn of the dialogues, in addition to the user utterances and system response, the annotations of belief state, database query results, and dialogue act are also provided. To process the data, we followed the approach used in UBAR [72]. Specifically, we employed delexicalization by replacing specific slot values with corresponding placeholders. These placeholders can be filled based on the results of a database search. The annotated dialogue acts serve as the stimulus in our approach. Table 4 provides information on all the dialogue acts and slots present in the dataset. We converted the structured dialogue acts, originally in the form of <domain, slot, value> triplets, into text format like *[domain1][inform] slot1 ... [request] slot1 ... [domain2][reqmore]*, where domains, acts, and slot values are all bracketed.

We used 780M Flan-T5-Large for our policy model to guide the ChatGPT and Codex LLMs. During the supervised fine-tuning of the policy model, we trained it to generate stimulus converted from the dialogue acts based on the given dialogue context. The policy model was trained for 25 epochs using 80 dialogues from the MultiWOZ2.0 and MultiWOZ2.1 datasets. When 800 dialogues are given, it was trained for 8 epochs on the MultiWOZ2.0 dataset and 20 epochs on the MultiWOZ2.1 dataset. All the hyperparameters setup is presented in Table 5.

Table 4: Full ontology for all domains in MultiWOZ2.0 [7] dataset. The upper script indicates which domains it belongs to. *: universal, 1: restaurant, 2: hotel, 3: attraction, 4: taxi, 5: train, 6: hospital, 7: police.

| dialogue acts | inform[*] / request[*] / select[1235] / recommend/[123] / nooffer[1235] / offerbook[125] / offerbooked[125] / nobook[12] / welcome[*] / greet[*] / bye[*] / reqmore[*] |
|---|---|
| slots | address[12367] / postcode[12367] / phone[123467] / name[123] / area[123] / pricerange[12] / type[23] / internet[2] / parking[2] / stars[2] / departure[45] / destination[45] / leave[45] / arrive[45] / people[123] / reference[1235] / id[5] / price[45] / time[15] / department[6] / day[125] / stay[2] / car[4] / food[1] |

| Model Params | Hyperparameter values |
|---|---|
| Supervised fine-tuning (SFT) | batch size: 8
epochs: $25/25/8/20$
learning rate: 0.00002
learning rate scheduler: linear
weight decay: 0.01 |
| RL (NLPO) | steps per update: 5120
total number of steps: 51200
batch size: 8
epochs per update: 5
learning rate: 0.000002
entropy coefficient: 0.0
initial kl coeff: 0.01
target kl: 0.2
discount factor: 0.99
gae lambda: 0.95
clip ratio: 0.2
value function coeff: 0.5
rollouts top k: 50
top mask ratio: 0.9
target update iterations: 20 |
| Tokenizer | padding side: left
truncation side: left
max length: 512 |
| Policy LM decoding | num_beams: 5
min length: 1
max new tokens: 40 |
| LLM decoding | sampling: True
temperature: 0.7
top_p: 1.0
max new tokens: 64 |

Table 5: Hyperparameters for experiments on the MultiWOZ dataset.

## A.3 Chain of Thought reasoning

We use our approach to generate instance-specific chain of thought (CoT) trigger prompts. Following previous work [26, 79], we evaluate two widely used arithmetic reasoning datasets MultiArith [57] and AQuA [35]. We compare with the 14 human-crafted chain-of-thought prompts evaluated in [26], part of which are collected from [1, 56]. We also compare with the prompt automatically designed by the APE approach [79]. We use the The hyperparameters used in our experiments are detailed in Table 6.

| Model Params | Hyperparameter values |
|---|---|
| Supervised fine-tuning (SFT) | batch size: 16
epochs: 2
learning rate: 0.00002
learning rate scheduler: linear
weight decay: 0.01 |
| RL (NLPO) | steps per update: 5120
total number of steps: 51200
batch size: 16
epochs per update: 5
learning rate: 0.000002
entropy coefficient: 0.0
initial kl coeff: 0.001
target kl: 0.5
discount factor: 0.99
gae lambda: 0.95
clip ratio: 0.2
value function coeff: 0.5
rollouts top k: 50
top mask ratio: 0.9
target update iterations: 20 |
| Tokenizer | padding side: right
truncation side: right
max length: 128 |
| Policy LM decoding | sampling: True
temperature: 0.7
max new tokens: 32 |
| LLM decoding | sampling: True
temperature: 0.7
top_p: 1.0
max new tokens: 32 |

Table 6: Hyperparameters for experiments on the zero-shot chain-of-thought reasoning.

# B  Additional results

## B.1  Summarization

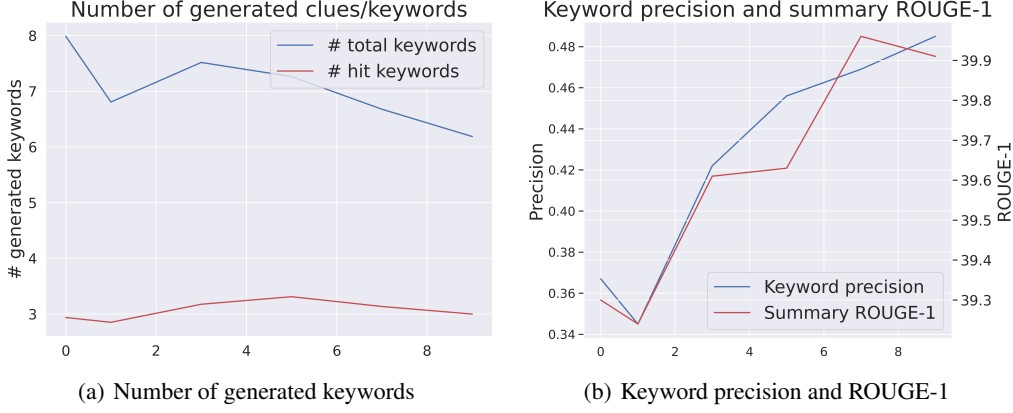

(a) Number of generated keywords

(b) Keyword precision and ROUGE-1

Figure 5: Number of generated keywords, keyword precision, and summary ROUGE-1 during the training process on 4000 samples.

**Analysis of generated hints/keywords**   We outlined changes in the number of generated keywords, hit keywords (those matched in the reference summary), and corresponding ROUGE-1 scores throughout the training process in Figure 5. As the training progresses, the policy model appears to generate keywords with increasing precision, which aligns positively with the increasing ROUGE-1 score. However, it is observed that even when keywords are generated with high precision if their quantity is too limited, the performance doesn't necessarily improve.

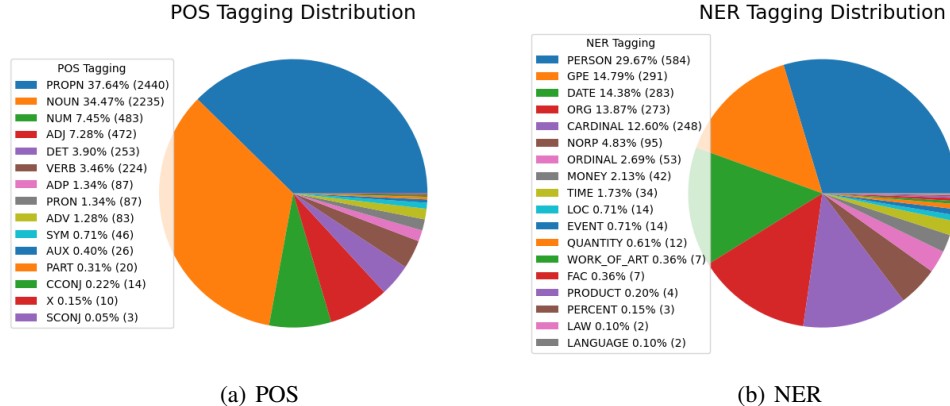

(a) POS                                          (b) NER

Figure 6: Part-of-Speech (POS) and Named Entity Recognition (NER) tagging on the generated hints, i.e., keywords.

We also employed the spacy package [20] for Part-of-Speech (POS) and Named Entity Recognition (NER) tagging on the generated keywords. The results are shown in the Figure 6. For the POS tagging, we observe that nouns (NOUN) and proper nouns (PROPN) are the most frequently generated keywords, which can serve as informative keywords. As for the NER tagging, the most commonly generated keywords include persons (PERSON), geopolitical entities (GPE), dates (DATE), organizations (ORG), and numerals (CARDINAL).

**GPT-4 Evaluation**    To gain a better understanding of generated summaries guided by keywords, we employed GPT-4 to evaluate the summaries. It has been shown that the LLM, especially GPT-4 is able to produce consistently high-quality assessments of text generation, showing high human alignment and thus being a good alternative to human evaluations [78, 38]. As we employ ROUGE scores as rewards for tuning the policy model to generate keywords that guide the LLM towards generating summaries more aligned with the reference summary, we leveraged GPT-4 to assess the overlap of key points (hints) between generated and reference summaries. Specifically, we use GPT-4 to compare the summaries generated with our proposed DSP and the original standard prompting. GPT-4 was instructed to first generate an explanation, followed by the corresponding answer. We prompt GPT-4 as follows:

> You are provided with an article and a corresponding reference summary. Additionally, there will be two alternative summaries labeled as 'A' and 'B'. Your task is to identify which of the two summaries (A or B) is more similar to the reference summary. This similarity should be evaluated based on the presence and accuracy of key points from the reference summary in each alternative summary. Please detail your reasoning in an explanation. After your explanation, classify the task outcome as: select 'A wins' if Summary A aligns more closely with the reference summary, 'B wins' if Summary B aligns more closely, or 'Tie' if both summaries align equally well with the reference summary.

The GPT-4 evaluation results are shown in Figure 7. We found that GPT-4 can produce reasonable and detailed explanations of their assessment. From our test set of 500 samples: DSP-generated summaries were favored 255 times (51.0%), summaries generated with original standard prompting were favored 222 times (44.4%), while a tie was observed in 23 cases (4.6%).

**Zero-shot prompting**    In our main experiments, we employ few-shot prompting with 3 examples in the prompt during training and evaluation. The specific prompt and demonstration examples utilized are detailed in Appendix D. To test whether the approach performed well under the zero-shot setting, we evaluated the following two experimental settings on the CNNDM dataset with 4,000 training samples: (1) few(3)-shot during training and zero-shot during evaluation; and (2) zero-shot during both training and evaluation.

As shown in Figure 8, when both training and testing are conducted using zero-shot prompting, the performance improvement over standard prompting is still comparable to the scenario where both are conducted using few-shot prompting. In addition, we observed that our approach exhibits

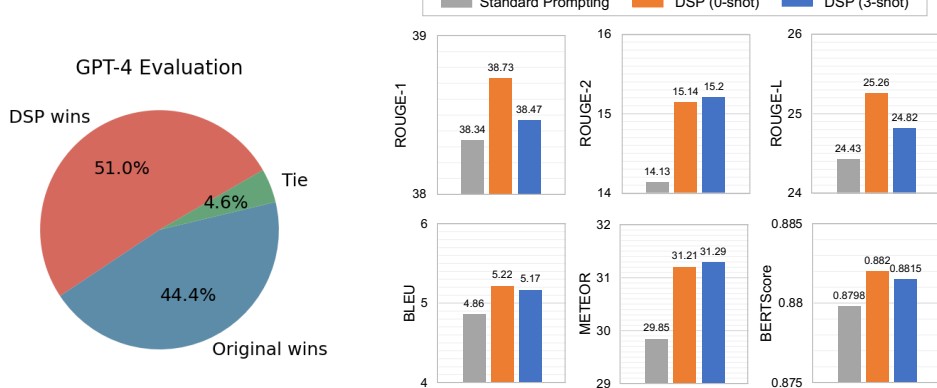

Figure 7: GPT-4 evaluation on comparing the summaries generated with our approach DSP, i.e., with the guidance of our generated keywords, and the original standard prompting, i.e., without keyword guidance.

Figure 8: Zero-shot evaluation results. DSP (0-shot) denotes that we use 0-shot prompting during RL training and DSP (3-shot) indicates we use 3-shot prompting during RL training.

Table 7: Low-resource evaluation on the MultiWOZ 2.0 dataset, where Succ. and Comb. denote the Success and Combined Score metrics, respectively.

| Method | 1% of training data (80 dialogues) | | | | 10% of training data (800 dialogues) | | | |
|---|---|---|---|---|---|---|---|---|
| | Inform | Succ. | BLEU | Comb. | Inform | Succ. | BLEU | Comb. |
| DAMD [77] | 34.4 | 9.1 | 8.1 | 29.9 | 55.3 | 30.3 | 13.0 | 55.8 |
| Soloist [49] | 58.4 | 35.3 | 10.6 | 57.4 | 69.9 | 51.9 | 14.6 | 75.5 |
| PPTOD [63] | 74.4 | 52.4 | 13.0 | 76.4 | 84.4 | 68.4 | 15.6 | 92.0 |
| UBAR [72] | - | - | - | - | 82.5 | 66.6 | 17.7 | 92.3 |
| GALAXY [19] | - | - | - | - | 90.0 | 75.9 | 17.5 | 100.2 |
| *Codex* | | | | | | | | |
| Standard Prompting | 76.7 | 41.5 | 7.7 | 66.8 | 76.7 | 41.5 | 7.7 | 66.8 |
| DSP w/ SFT | 74.9 | 66.3 | 11.1 | 81.7 | 79.4 | 71.9 | 11.3 | 87.0 |
| DSP w/ SFT+RL | 91.0 | 76.0 | 9.8 | 93.3 | 96.0 | 86.9 | 10.7 | 102.2 |
| *ChatGPT* | | | | | | | | |
| Standard Prompting | 71.8 | 44.1 | 10.5 | 68.4 | 71.8 | 44.1 | 10.5 | 68.4 |
| DSP w/ SFT | 76.6 | 66.5 | 11.2 | 82.8 | 72.7 | 64.7 | 11.8 | 80.5 |
| DSP w/ SFT+RL | 90.9 | 82.2 | 10.2 | 96.7 | 95.3 | 82.3 | 10.9 | 99.6 |

robustness when different numbers of examples are used in prompts during training and evaluation, as our approach with few(3)-shot training still outperforms standard prompting under zero-shot testing.

## B.2 Dialogue response generation

**Low-resource results** In addition to the performance of compared baseline models with full training data as shown in the main paper, we also present their performance in the low-resource setting in Table 7. It is important to note that most of these methods struggle to achieve acceptable performance with only 1% of the training data (80 dialogues), and thus their results in the 1% setting are not reported. As for those with reported performance with 80 dialogues, their results are significantly worse compared to Codex and ChatGPT guided by the policy model. Furthermore, even with around 800 dialogues, their Inform and Success rates were still much lower than those achieved by ChatGPT and Codex.

### B.3 Chain-of-Thought reasoning

Table 8: Some generated trigger prompts by our fine-tuned policy model.

| Generated CoT Trigger Prompts |
| --- |
| *First step:* |
| *Let's think like a detective step by step. First,* |
| *Let's solve this problem by splitting it into steps. First,* |
| *Let's think step by step using common sense.* |
| *Let's think step by step using our creative brains.* |
| *Let's think step by step using both the above information and the testing.* |
| *Let's think step by step using proven methods.* |
| *The answer is following the proof.* |

**Newly discovered prompts** After fine-tuning with RL, the policy model is encouraged to discover instance-specific trigger prompts. Table 8 showcases some of these newly generated trigger prompts, which deviate from those present in the training data for SFT. Some are modifications or combinations of prompts from the training data, such as "*First step:*" and "*Let's think like a detective step by step. First,*". Others include new information, like "*Let's think step by step using both the above information and the testing.*" and "*Let's think step by step using proven methods.*".

## C  Running examples

We provide two running examples on the CNN/Daily Mail and MultiWOZ dataset in Table 9 and 10, respectively. For each example, we present the generations of ChatGPT with standard prompting, DSP trained with SFT, and DSP trained with SFT and RL.

## D  Prompts

The used prompts of standard prompting and our proposed Directional Stimulus Prompting on CNN/Daily Mail and MultiWOZ datasets are given in Figures 9, 10, and Figures 11, 12, respectively. Both use the same three demonstration examples in standard prompting and DSP. In the case of the CNN/Daily Mail dataset, DSP incorporates additional keywords as hints (stimulus) in the prompts. For the MultiWOZ dataset, DSP includes the dialogue acts for each system turn as stimulus, along with explanations for all the dialogue acts.

| Input article | The winter of 2014-15 won't be easily forgotten in Boston after the endless snow broke countless records and the city had to pay volunteers $30 an hour to help dig out the battered city. The shere volume of snow that fell earlier this year, nearly 65 inches fell in February alone, means that huge piles of the white stuff still remain. Except the remaining 'snow' isn't very white any more but rather a disgusting black color riddled with trash including broken pieces of glass, plastic shards and goodness knows what else. Scroll down for video . Vlad Tarasov couldn't resist filming himself ski down the slopes at Boston's largest snow farm located in the city's Seaport District . The one-minute video gives a first-person perspective of pushing through the filthy, trash-filled ice pile that served as a dumping ground for the snow . To some avid skiers snow is still snow and one in particular couldn't resist the urge to take to the slopes of Boston's temporary new resort. Vlad Tarasov even filmed his journey down the slopes at Boston's largest snow farm located in the city's Seaport District. 'I've been skiing for 20 years, but never like this,' he told The Boston Globe about the 'surreal' experience of climbing the slopes on April 5 and looking down the South Boston urban sprawl. The one-minute video gives viewers a first-person perspective of the experience as Tarasov pushes through the filthy, trash-filled ice pile that served as a dumping ground for the historic winter snowfall. Tarasov recalls having to avoid junk including rusted lawn chairs, parking cones, broken bottles, and 'pretty much every kind of trash you'd ever find on a Boston street.' He hit a dead seagull, he said. 'I've been skiing for 20 years, but never like this,' said Tarasov about the 'surreal' experience . Junk in the filthy snow included rusted lawn chairs, parking cones, broken bottles and even a dead seagull . 'I'm so accustomed to skiing on the white stuff. I was definitely concerned once I climbed the mound about falling on debris,' said Tarasov. He has no plans to repeat his daredevil stunt as the warm weather is now finally melting the snow farm and he doesn't want to have to clean all his gear again. Boston received 108.6 inches of snow - about 9 feet - during this last winter, beating the previous seasonal record of 107.9 inches that was set in 1995-96. The 65 inches that fell in February crushed the previous one-month record of 43.3 inches in January 2005. The city of Boston received 108.6 inches of snow - about 9 feet - during this last winter, topping a seasonal record of 107.9 inches that was set in 1995-96 . |
|---|---|
| Reference summary | Vlad Tarasov couldn't resist filming himself ski down the slopes at Boston's largest snow farm located in the city's Seaport District . His one-minute video gives a first-person perspective of pushing through the filthy, trash-filled ice pile that served as a dumping ground for the snow . 'I've been skiing for 20 years, but never like this,' he said about the 'surreal' experience . Junk in the filthy snow included rusted lawn chairs, parking cones, broken bottles and even a dead seagull . |

| *Standard prompting* | |
|---|---|
| Generated summary (ChatGPT) | A video has emerged of a skier named Vlad Tarasov skiing on a pile of snow in the Seaport District of Boston. The pile of snow was where the city put the snow that fell during the record-breaking winter season. Unfortunately, the snow is now a disgusting black color riddled with trash, including broken pieces of glass, plastic shards and other debris. |
| ROUGE-1 | 37.58 |

| *Directional Stimulus Prompting (SFT)* | |
|---|---|
| Generated keywords (T5) | Vlad Tarasov; 65 inches; Boston; Seaport District; the city; $30 an hour; volunteers. |
| Generated summary (ChatGPT) | Vlad Tarasov filmed himself skiing down the slopes of Boston's largest snow farm located in the city's Seaport District, which served as a dumping ground for the historic winter snowfall . The shere volume of snow that fell earlier this year, nearly 65 inches fell in February alone, means that huge piles of the white stuff still remain and the city had to pay volunteers $30 an hour to help dig out the battered city. |
| ROUGE-1 | 46.91 |

| *Directional Stimulus Prompting (SFT+RL)* | |
|---|---|
| Generated keywords (T5) | Vlad Tarasov; ski; Boston; large snow farm; Seaport District; surreal; video; filthy; snowfall. |
| Generated summary (ChatGPT) | Vlad Tarasov couldn't resist filming himself skiing down the slopes of Boston's largest snow farm located in the city's Seaport District . To some avid skiers, snow is still snow and Tarasov said it was a 'surreal' experience. The one-minute video gives viewers a first-person perspective of the experience as Tarasov pushes through the filthy, trash-filled ice pile that served as a dumping ground for the historic winter snowfall. |
| ROUGE-1 | 71.25 |

Table 9: An example from the CNN/Daily Mail dataset. The policy model (T5) generates the keywords based on which the ChatGPT generates the summary. The keywords that are mentioned in the reference summary are underlined and highlighted in red.

| Dialogue context | **User:** i want to find a moderate -ly priced restaurant .
**Assistant:** i have [value_choice] available for you ! is there a certain area or cuisine that interests you ?
**User:** yes i would like the restaurant to be located in the center of the attractions . [db_3] |
|---|---|
| **Oracle dialogue acts** | [restaurant] [inform] choice [request] food |
| **Oracle system response** | there are [value_choice] restaurant -s available in the [value_area] . how about a specific type of cuisine ? |
| *Standard prompting* | |
| **Generated response (Chat-GPT)** | the [value_name] is located in the center of the attractions . would you like me to book a table ? |
| **SacreBLEU** | 14.09 |
| *Directional Stimulus Prompting (SFT)* | |
| **Generated dialogue acts (T5)** | [restaurant] [inform] food name area choice [recommend] price |
| **Generated response (Chat-GPT)** | i would suggest [value_name] in the [value_area] area . they have [value_food] cuisine and are moderate in price . are you interested ? |
| **SacreBLEU** | 16.40 |
| *Directional Stimulus Prompting (SFT+RL)* | |
| **Generated dialogue acts (T5)** | [restaurant] [inform] choice [request] food |
| **Generated summary (ChatGPT)** | i have [value_choice] restaurants in the area . do you have a specific cuisine in mind ? |
| **SacreBLEU** | 22.80 |

Table 10: An example from the MultiWOZ dataset. The policy model (T5) generates the dialogue acts given the dialog context. With our approach DSP, ChatGPT generates the response conditioned on the generated dialogue acts.

Given an article, write a short summary in 2-4 sentence.

Article: Seoul (CNN) South Korea's Prime Minister Lee Wan-koo offered to resign on Monday amid a growing political scandal. Lee will stay in his official role until South Korean President Park Geun-hye accepts his resignation. He has transferred his role of chairing Cabinet meetings to the deputy prime minister for the time being, according to his office. Park heard about the resignation and called it "regrettable," according to the South Korean presidential office. Calls for Lee to resign began after South Korean tycoon Sung Woan-jong was found hanging from a tree in Seoul in an apparent suicide on April 9. Sung, who was under investigation for fraud and bribery, left a note listing names and amounts of cash given to top officials, including those who work for the President. Lee and seven other politicians with links to the South Korean President are under investigation. A special prosecutor's team has been established to investigate the case. Lee had adamantly denied the allegations as the scandal escalated: "If there are any evidence, I will give out my life. As a Prime Minister, I will accept Prosecutor Office's investigation first." Park has said that she is taking the accusations very seriously. Before departing on her trip to Central and South America, she condemned political corruption in her country. "Corruption and deep-rooted evil are issues that can lead to taking away people's lives. We take this very seriously." "We must make sure to set straight this issue as a matter of political reform. I will not forgive anyone who is responsible for corruption or wrongdoing." Park is in Peru and is expected to arrive back to South Korea on April 27. CNN's Paula Hancocks contributed to this report.
Q: Write a short summary of the article in 2-4 sentences.
A: Calls for Lee Wan-koo to resign began after South Korean tycoon Sung Woan-jong was found hanging from a tree in Seoul . Sung, who was under investigation for fraud and bribery, left a note listing names and amounts of cash given to top officials .

Article: The BBC has refused to hand over the emails of a deceased woman to her grieving husband, who believes they will prove she was 'bullied' by the Corporation's management towards the end of her life. Mother-of-two Marie Csaszar, 45, died last September following a ten-year battle with a brain tumour. She had worked for seven years at the BBC's financial centre in Cardiff as a contracts manager, but according to her husband Paul, she was forced out of the post into another job after drawing attention to management blunders which he says cost licence-fee payers about £150,000. Whistleblower: Marie Csaszar died after a 10-year battle with a brain tumour. Her widower has had a request for her work emails, which he believes will show she was being bullied by bosses, refused by the BBC . Legal experts described the case as 'highly unusual', but predicted that it could be followed by similar claims as digital documents such as emails and social media posts play an increasingly important part in people's lives. Facebook users in the US have the chance to designate a 'legacy contact' who can take over parts of their account after their death. Mr Csaszar says he believes the BBC failed in its duty of care to his late wife, and the treatment she received from management affected her health. He asked the BBC under the Data Protection Act for copies of his wife's emails, in the hope they will provide evidence of her 'appalling' treatment by the Corporation, which he has spent months pursuing. However, he was told last month by the BBC that under the Act, personal data is defined as only 'data which relates to a living individual'. Despite Mr Csaszar being his wife's legal next of kin, an email from the BBC – seen by The Mail on Sunday – read: 'Unfortunately as your wife has sadly passed away it is not possible for you to exercise her right to access her personal data under the Data Protection Act.' Mr Csaszar said his wife gave evidence to the BBC's 2013 Respect At Work review that was set up to look into sexism and bullying following the Jimmy Savile scandal. He said it was her evidence which led the chairwoman, Dinah Rose QC, to conclude some senior Corporation staff were viewed as 'untouchable'. Details of the individuals who were the subject of Mr Csaszar's complaint cannot be published for legal reasons but they involve at least one senior manager on a six-figure salary. Last night, self-employed businessman Mr Csaszar, 48, said: 'Marie was being hammered from one side to the other towards the end of her life. Nobody stood up and objected. Locked out: The BBC has refused to hand over the work emails of Marie Csaszar to her widower Paul . 'The treatment which Marie experienced at the hands of the BBC affected her recovery. She was constantly upset and left in tears, saying, "I don't know what I've done to deserve being treated like this." 'They didn't want Marie on the staff because she knew too much about what had gone on. 'She would refuse to sign off certain contracts because they didn't show value for money, but they were signed off anyway. This caused her a great deal of stress.' Mrs Csaszar underwent intensive treatment for her cancer over several years, which left her suffering with epilepsy. However, she continued to work in her £50,000-a-year post, and raised serious concerns that could have saved licence-fee payers' money, her husband claims. In a formal grievance procedure with the BBC in 2013, Mrs Csaszar raised a complaint about the closure of a cash office run by an outside contractor. She was advised by the BBC's legal department that the Corporation was not liable under its contract to pay any money towards redundancy for staff employed by the contractors. But she told her husband that about £35,000 was paid out anyway. Mr Csaszar says he is desperate to see emails referring to his wife and how she was treated following her whistleblowing. He contacted the Information Commissioner's Office, only to be told that the release of the documents was 'at the discretion' of the BBC. Mr Csaszar has raised his request with the BBC Trust, the Corporation's watchdog, and is considering legal action. A BBC spokesman said: 'We were very sorry to hear of Ms Csaszar's death. A thorough investigation was carried out into her complaint and the outcome was provided to her in July 2013. We have provided this response to her husband. Ms Csaszar then attended work from October 2013 to May 2014 and did not raise a further appeal to her complaint.'
Q: Write a short summary of the article in 2-4 sentences.
A: Mother-of-two died following a 10-year battle with a brain tumour . She worked at BBC in Cardiff where husband claims she was bullied . Marie Csaszar gave evidence at BBC's 2013 Respect At Work review . He asked for emails under the Data Protection Act but was refused .

Article: (CNN) An American citizen was wounded by gunfire Thursday as she drove from the medical school in Karachi, Pakistan, where she works, police said. Debra Lobo, a 55-year-old California native, was shot in the right cheek and left arm and is unconscious but expected to survive, according to Mohamad Shah, a Karachi police spokesman. Police found pamphlets that the assailants had thrown into Lobo's car, written in Urdu, saying "America should be burnt," Shah said. Lobo had left the Jinnah Medical and Dental College, where she works as vice principal, to pick up her two daughters from school. Two assailants on a passing motorcycle shot her while she was driving, Shah said. "Our U.S. Consulate General in Karachi is in close contact with Pakistani authorities and is working to obtain more information," said a U.S. Embassy spokesperson. Lobo is being treated at the Karachi's Aga Khan Hospital, said Shah. She has lived in Pakistan since 1996 and is married to a Christian Pakistani who is a librarian at the American School in Karachi. Karachi police are investigating, Shah said.
Q: Write a short summary of the article in 2-4 sentences.
A: Debra Lobo, 55, is unconscious but is expected to survive after being shot Thursday, police say . She is vice principal of the Jinnah Medical and Dental College in Karachi . Police: She was on her way to pick up her daughters from school when she was shot .

Article: [[QUESTION]]
Q: Write a short summary of the article in 2-4 sentences.
A:

Figure 9: The prompt for standard prompting on the CNN/Daily Mail dataset.

## Directional Stimulus Prompt (CNN/Daily Mail)

Given an article and a list of keywords, write a short summary that accurately incorporates the provided keywords into 2-4 sentences.

Article: Seoul (CNN) South Korea's Prime Minister Lee Wan-koo offered to resign on Monday amid a growing political scandal. Lee will stay in his official role until South Korean President Park Geun-hye accepts his resignation. He has transferred his role of chairing Cabinet meetings to the deputy prime minister for the time being, according to his office. Park heard about the resignation and called it "regrettable," according to the South Korean presidential office. Calls for Lee to resign began after South Korean tycoon Sung Woan-jong was found hanging from a tree in Seoul in an apparent suicide on April 9. Sung, who was under investigation for fraud and bribery, left a note listing names and amounts of cash given to top officials, including those who work for the President. Lee and seven other politicians with links to the South Korean President are under investigation. A special prosecutor's team has been established to investigate the case. Lee had adamantly denied the allegations as the scandal escalated: "If there are any evidence, I will give out my life. As a Prime Minister, I will accept Prosecutor Office's investigation first." Park has said that she is taking the accusations very seriously. Before departing on her trip to Central and South America, she condemned political corruption in her country. "Corruption and deep-rooted evil are issues that can lead to taking away people's lives. We take this very seriously." "We must make sure to set straight this issue as a matter of political reform. I will not forgive anyone who is responsible for corruption or wrongdoing." Park is in Peru and is expected to arrive back to South Korea on April 27. CNN's Paula Hancocks contributed to this report.
Q: Write a short summary of the article in 2-4 sentences that accurately incorporates the provided keywords.
Keywords: Lee Wan-koo; resign, South Korean tycoon; Sung Woan-jong; hanging from a tree; investigation; notes; top officials.
A: Calls for Lee Wan-koo to resign began after South Korean tycoon Sung Woan-jong was found hanging from a tree in Seoul . Sung, who was under investigation for fraud and bribery, left a note listing names and amounts of cash given to top officials .

Article: The BBC has refused to hand over the emails of a deceased woman to her grieving husband, who believes they will prove she was 'bullied' by the Corporation's management towards the end of her life. Mother-of-two Marie Csaszar, 45, died last September following a ten-year battle with a brain tumour. She had worked for seven years at the BBC's financial centre in Cardiff as a contracts manager, but according to her husband Paul, she was forced out of the post into another job after drawing attention to management blunders which he says cost licence-fee payers about £150,000. Whistleblower: Marie Csaszar died after a 10-year battle with a brain tumour. Her widower has had a request for her work emails, which he believes will show she was being bullied by bosses, refused by the BBC . Legal experts described the case as 'highly unusual', but predicted that it could be followed by similar claims as digital documents such as emails and social media posts play an increasingly important part in people's lives. Facebook users in the US have the chance to designate a 'legacy contact' who can take over parts of their account after their death. Mr Csaszar says he believes the BBC failed in its duty of care to his late wife, and the treatment she received from management affected her health. He asked the BBC under the Data Protection Act for copies of his wife's emails, in the hope they will provide evidence of her 'appalling' treatment by the Corporation, which he has spent months pursuing. However, he was told last month by the BBC that under the Act, personal data is defined as only 'data which relates to a living individual'. Despite Mr Csaszar being his wife's legal next of kin, an email from the BBC – seen by The Mail on Sunday – read: 'Unfortunately as your wife has sadly passed away it is not possible for you to exercise the right to access her personal data under the Data Protection Act.' Mr Csaszar said his wife gave evidence to the BBC's 2013 Respect At Work review that was set up to look into sexism and bullying following the Jimmy Savile scandal. He said it was her evidence which led the chairwoman, Dinah Rose QC, to conclude some senior Corporation staff were viewed as 'untouchable'. Details of the individuals who were the subject of Mr Csaszar's complaint cannot be published for legal reasons but they involve at least one senior manager on a six-figure salary. Last night, self-employed businessman Mr Csaszar, 48, said: 'Marie was being hammered from one side to the other towards the end of her life. Nobody stood up and objected. Locked out: The BBC has refused to hand over the work emails of Marie Csaszar to her widower Paul . 'The treatment which Marie experienced at the hands of the BBC affected her recovery. She was constantly upset and left in tears, saying, "I don't know what I've done to deserve being treated like this." 'They didn't want Marie on the staff because she knew too much about what had gone on. 'She would refuse to sign off certain contracts because they didn't show value for money, but they were signed off anyway. This caused her a great deal of stress.' Mrs Csaszar underwent intensive treatment for her cancer over several years, which left her suffering with epilepsy. However, she continued to work in her £50,000-a-year post, and raised serious concerns that could have saved licence-fee payers' money, her husband claims. In a formal grievance procedure with the BBC in 2013, Mrs Csaszar raised a complaint about the closure of a cash office run by an outside contractor. She was advised by the BBC's legal department that the Corporation was not liable under its contract to pay any money towards redundancy for staff employed by the contractors. But she told her husband that about £35,000 was paid out anyway. Mr Csaszar says he is desperate to see emails referring to his wife and how she was treated following her whistleblowing. He contacted the Information Commissioner's Office, only to be told that the release of the documents was 'at the discretion' of the BBC. Mr Csaszar has raised his request with the BBC Trust, the Corporation's watchdog, and is considering legal action. A BBC spokesman said: 'We were very sorry to hear of Ms Csaszar's death. A thorough investigation was carried out into her complaint and the outcome was provided to her in July 2013. We have provided this response to her husband. Ms Csaszar then attended work from October 2013 to May 2014 and did not raise a further appeal to her complaint.'
Q: Write a short summary of the article in 2-4 sentences that accurately incorporates the provided keywords.
Keywords: Mother; battle; brain tumour; BBC; Cardiff; husband; bullied; Marie Csaszar; evidence; Respect At Work; emails; Data Protection Act.
A: Mother-of-two died following a 10-year battle with a brain tumour . She worked at BBC in Cardiff where husband claims she was bullied . Marie Csaszar gave evidence at BBC's 2013 Respect At Work review . He asked for emails under the Data Protection Act but was refused .

Article: (CNN) An American citizen was wounded by gunfire Thursday as she drove from the medical school in Karachi, Pakistan, where she works, police said. Debra Lobo, a 55-year-old California native, was shot in the right cheek and left arm and is unconscious but expected to survive, according to Mohamad Shah, a Karachi police spokesman. Police found pamphlets that the assailants had thrown into Lobo's car, written in Urdu, saying "America should be burnt," Shah said. Lobo had left the Jinnah Medical and Dental College, where she works as vice principal, to pick up her two daughters from school. Two assailants on a passing motorcycle shot her while she was driving, Shah said. "Our U.S. Consulate General in Karachi is in close contact with Pakistani authorities and is working to obtain more information," said a U.S. Embassy spokesperson. Lobo is being treated at the Karachi's Aga Khan Hospital, said Shah. She has lived in Pakistan since 1996 and is married to a Christian Pakistani who is a librarian at the American School in Karachi. Karachi police are investigating, Shah said.
Q: Write a short summary of the article in 2-4 sentences that accurately incorporates the provided keywords.
Keywords: Debra Lobo; 55; unconscious; Thursday; vice principal; the Jinnah Medical and Dental College; Karachi; daughters; school.
A: Debra Lobo, 55, is unconscious but is expected to survive after being shot Thursday, police say . She is vice principal of the Jinnah Medical and Dental College in Karachi . Police: She was on her way to pick up her daughters from school when she was shot .

Article: [[QUESTION]]
Q: Write a short summary of the article in 2-4 sentences that accurately incorporates the provided keywords.
Keywords: [[HINT]]
A:

Figure 10: The prompt for Directional Stimulus Prompting on the CNN/Daily Mail dataset. The difference compared with the prompts used in standard prompting shown in Figure 9 is the stimulus hints (keywords), which are highlighted in red.

## Standard Prompt (MultiWOZ)

The following are conversations between a user and an assistant. The assistant can help the user with booking or providing information of restaurant, hotel, attraction, taxi, train, hospital, police, and so on.

Conversation:
User: do you book trains for the bishops stortford ?
Assistant: yes . are you departing or arriving in [value_destination] ?
User: i am going there from cambridge on thursday .
Assistant: i have [value_choice] trains going to [value_destination] from [value_departure] on [value_day] . what time do you need to arrive ?
User: i would like to arrive by 18:15 .
Assistant: [value_id] will have you in [value_destination] by [value_arrive] .
User: i need 2 tickets .
Assistant: the [value_id] will get you there by [value_arrive] . i can book you [value_people] tickets on that if you would like ?
User: 2 tickets will be fine , please .
Assistant: tr0277 2 tickets . the total fee is [value_price] payable at the station . reference number is : [value_reference] . will there be anything else i can help you with today ?
User: a nice place to eat that serves austrian food . in the centre of town if possible .
Assistant: i am so sorry no [value_food] restaurant -s in that area , another food type perhaps ?
User: okay , do you have any good restaurant -s serving british food ?
Assistant: there are [value_choice] restaurant -s in the [value_area] that serve [value_food] food , is there a price range you would like to stay in ?
User: any is fine , can i get the phone number and postcode of 1 you would recommend ?
Assistant: i would recommend the [value_name] . their phone number is [value_phone] . their postcode is [value_postcode] .
User: great ! thanks . that s all i need . goodbye .
Assistant: awesome . we 're here 24 and 7 if you need us again . have a great day !

Conversation:
User: i ' m looking for a hotel in the moderate price range . i definitely want a hotel , not a guest house .
Assistant: of course . there are [value_choice] [value_type] in the [value_area] and [value_choice] in the [value_area] of town . do you have a preference ?
User: is there anything to the west , preferably with a 4 star rating ?
Assistant: i ' m sorry but nothing exists in [value_area] north with [value_stars] 4 star rateing in [value_price] moderate price range . could we try another area ?
User: is there 1 in the cheap range ?
Assistant: no , there is not .
User: oh goodness . can you check once again for a hotel , star 4 , in the west . cheap is fine .
Assistant: the [value_name] is located at [value_address] . do you need more information or would you like to book now ?
User: yeah book for 5 people .
Assistant: for how many days ?
User: will this start on friday ?
Assistant: yes . would you like reservations ? what day will it start ? how many people ?
User: i would like a reservation for 5 people for 4 nights , beginning friday , at the cambridge belfry .
Assistant: you are all booked and your reservation number is [value_reference] .
User: thank you for the reservation . i am also looking for a lovely park in north cambridge .
Assistant: i would suggest [value_name] .
User: what is the phone number ?
Assistant: the phone number is [value_phone] .
User: i also need a taxi between the 2 place -s .
Assistant: please give me the travel time , departure location and arrival location .
User: i want to leave the hotel by 04:30 .
Assistant: what time do you need to leave the restaurant by ?
User: i want to be pickup at at the hotel by 04:30 .
Assistant: great your booking is complete . you will be picked up in a [value_car] by [value_leave] at [value_departure] and dropped off at the [value_destination] .
User: do you have their contact number ?
Assistant: their contact number is [value_phone] . is there anything else i can help you with today ?
User: that will be all for today . thank you . goodbye .
Assistant: thank you i ' m glad i could help . goodbye .

Conversation:
User: i want to find place -s to go in the centre of town that are park type attractions .
Assistant: i think that you will like [value_name] . it is [value_price] , may i book it for you ?
User: yes , that sounds good ! please provide me with the postcode .
Assistant: i ' m sorry , we are n't able to book tickets for this attraction , but their postcode [value_postcode] cb21jf , and i can provide their address and phone if you like ?
User: no . that is all i need . i would like help finding a cheap guest house though .
Assistant: i have [value_choice] options for you . do you need wifi or parking ? is there a part of town you prefer to stay in ?
User: it should include free wifi and should be in the east .
Assistant: there seems to be an opening at [value_name] . would you like me to book you a room ?
User: yes , please . 5 people for 5 nights starting thursday .
Assistant: booking was successful . your reference number is : [value_reference] .
User: thank you very much !
Assistant: you are welcome . is there anything else i can do for you ?
User: no thank you . i appreciate all of your help .
Assistant: thank you i ' m glad i could help .

Conversation:
[[DIALOG]]

Figure 11: The prompt for standard prompting on the MultiWOZ dataset.

## Directional Stimulus Prompt (MultiWOZ)

The following are conversations between a user and an assistant. Indicated by the dialog acts, the assistant can help the user with booking or providing information of [restaurant], [hotel], [attraction], [taxi], [train], [hospital], [police], and so on.

**Dialog acts:**
**[inform]: provide information for a slot; [request]: request information for a slot; [nooffer]: do not have offers that satisfy requirements; [recommend]: recommend an offer; [select]: select an offer; [offerbook]: help to book a [offerbooked]: have booked an offer; [nobook]: can not book an offer; [bye]: bye; [greet]: greet; [welcome]: welcome; [reqmore]: ask if need more information or help.**

Conversation:
User: do you book trains for the bishops stortford ?
Assistant(**[train] [request] departure destination**): yes . are you departing or arriving in [value_destination] ?
User: i am going there from cambridge on thursday .
Assistant([train] [inform] day choice departure destination [request] arrive): i have [value_choice] trains going to [value_destination] from [value_departure] on [value_day] . what time do you need to arrive ?
User: i would like to arrive by 18:15 .
Assistant(**[train] [inform] id destination arrive):** [value_id] will have you in [value_destination] by [value_arrive] .
User: i need 2 tickets .
Assistant(**[train] [inform] arrive id [offerbook] people):** the [value_id] will get you there by [value_arrive] . i can book you [value_people] tickets on that if you would like ?
User: 2 tickets will be fine , please .
Assistant(**[train] [offerbooked] reference price [general] [reqmore]):** tr0277 2 tickets . the total fee is [value_price] payable at the station . reference number is : [value_reference] . will there be anything else i can help you with today ?
User: a nice place to eat that serves austrian food . in the centre of town if possible .
Assistant(**[restaurant] [nooffer] food [request] food):** i am so sorry no [value_food] restaurant -s in that area , another food type perhaps ?
User: okay , do you have any good restaurant -s serving british food ?
Assistant(**[restaurant] [inform] choice food area [request] price):** there are [value_choice] restaurant -s in the [value_area] that serve [value_food] food , is there a price range you would like to stay in ?
User: any is fine , can i get the phone number and postcode of 1 you would recommend ?
Assistant(**[restaurant] [recommend] postcode phone name):** i would recommend the [value_name] . their phone number is [value_phone] . their postcode is [value_postcode] .
User: great ! thanks . that s all i need . goodbye .
Assistant(**[general] [bye]):** awesome . we 're here 24 and 7 if you need us again . have a great day !

Conversation:
User: i ' m looking for a hotel in the moderate price range . i definitely want a hotel , not a guest house .
Assistant(**[hotel] [inform] area choice type [request] area):** of course . there are [value_choice] [value_type] in the [value_area] and [value_choice] in the [value_area] of town . do you have a preference ?
User: is there anything to the west , preferably with a 4 star rating ?
Assistant(**[hotel] [nooffer] area price stars [request] area):** i ' m sorry but nothing exists in [value_area] north with [value_stars] 4 star rateing in [value_price] moderate price range . could we try another area ?
User: is there 1 in the cheap range ?
Assistant(**[hotel] [nooffer]):** no , there is not .
User: oh goodness . can you check once again for a hotel , star 4 , in the west . cheap is fine .
Assistant(**[hotel] [inform] name address [offerbook] [general] [reqmore]):** the [value_name] is located at [value_address] . do you need more information or would you like to book now ?
User: yeah book for 5 people .
Assistant(**[hotel] [request] stay):** for how many days ?
User: will this start on friday ?
Assistant(**[hotel] [offerbook] [request] people day):** yes . would you like reservations ? what day will it start ? how many people ?
User: i would like a reservation for 5 people for 4 nights , beginning friday , at the cambridge belfry .
Assistant(**[hotel] [offerbooked] reference):** you are all booked and your reservation number is [value_reference].
User: thank you for the reservation . i am also looking for a lovely park in north cambridge .
Assistant(**[attraction] [recommend] name):** i would suggest [value_name] .
User: what is the phone number ?
Assistant(**[attraction] [inform] phone):** the phone number is [value_phone] .
User: i also need a taxi between the 2 place -s .
Assistant(**[taxi] [request] departure leave destination):** please give me the travel time , departure location and arrival location .
User: i want to leave the hotel by 04:30 .
Assistant(**[taxi] [request] leave):** what time do you need to leave the restaurant by ?
User: i want to be pickup at at the hotel by 04:30 .
Assistant(**[taxi] [inform] departure car leave destination):** great your booking is complete . you will be picked up in a [value_car] by [value_leave] at [value_departure] and dropped off at the [value_destination] .
User: do you have their contact number ?
Assistant(**[taxi] [inform] phone [general] [reqmore]):** their contact number is [value_phone] . is there anything else i can help you with today ?
User: that will be all for today . thank you . goodbye .
Assistant(**[general] [bye]):** have a great day ! bye .

Conversation:
User: i want to find place -s to go in the centre of town that are park type attractions .
Assistant(**[attraction] [inform] price name [offerbook]):** i think that you will like [value_name] . it is [value_price] , may i book it for you ?
User: yes , that sounds good ! please provide me with the postcode .
Assistant(**[attraction] [nobook] [inform] postcode):** i ' m sorry , we are n't able to book tickets for this attraction , but their postcode [value_postcode] cb21jf , and i can provide their address and phone if you like ?
User: no . that is all i need . i would like help finding a cheap guest house though .
Assistant(**[hotel] [inform] choice [request] area parking internet):** i have [value_choice] options for you . do you need wifi or parking ? is there a part of town you prefer to stay in ?
User: it should include free wifi and should be in the east .
Assistant(**[hotel] [inform] name [offerbook]):** there seems to be an opening at [value_name] . would you like me to book you a room ?
User: yes , please . 5 people for 5 nights starting thursday .
Assistant(**[hotel] [offerbooked] reference):** booking was successful . your reference number is : [value_reference] .
User: thank you very much !
Assistant(**[general] [welcome] [reqmore]):** you are welcome . is there anything else i can do for you ?
User: no thank you . i appreciate all of your help .
Assistant(**[general] [welcome]):** thank you i ' m glad i could help .

Conversation:
[[DIALOG]]

Figure 12: The prompt for Directional Stimulus Prompting on the MultiWOZ. Compared with the prompts used in standard prompting shown in Figure 9, we add stimulus hints (dialogue acts) for each system turn, which are highlighted in red. In addition, we add explanations of dialogue acts at the beginning to help the model understand their meanings.

