# OpenReview forum: "Guiding Large Language Models via Directional Stimulus Prompting"
_NeurIPS.cc/2023/Conference — NeurIPS 2023 poster_

### Official Review · Reviewer_2Z7n · 2023-06-24

**Soundness:** 3 good
**Presentation:** 4 excellent
**Contribution:** 2 fair
**Rating:** 4
**Confidence:** 4

**Summary:**

This paper proposes a Directional Stimulus Prompting method to provide fine-grained guidance for the output of large language models (LLMs). This method introduces a small trainable policy model (e.g. FLAN-T5) to generate hints for each query to guide LLMs towards desired outputs. This policy model can be trained via a standard paradigm including supervised fine-tuning on pseudo-labeled data and reinforcement learning (RL) to optimize the expected reward. Experiments on summarization and dialogue generation tasks show the effectiveness of the proposed method.

**Strengths:**

1. This paper proposes a direct and feasible solution to guide black-box LLMs towards better generation results, which is widely applicable with the rapid development of LLMs.
2. This paper is well-written and easy to follow.


**Weaknesses:**

1. The technical novelty of the proposed method is limited. The training of the policy model in this paper is a standard paradigm including supervised fine-tuning and reinforcement learning. The techniques used in RL including dynamic adjustment of the coefficient and NLPO are all borrowed from existing works. Thus, I feel that the main difference falls into the usage of LLMs as an evaluation function in the reward model, which is devised for adapting this RL algorithm to the scenario of guiding LLM. Thus, the novelty of this design is mainly on the applicational side.

2. During supervised fine-tuning, the authors heuristically select the “pseudo-stimulus” for each input. The pseudo-stimulus indicates keywords / dialogue acts for summarization / task-oriented dialogue generation, respectively. I wonder whether there is a general principle to select pseudo-stimulus especially in open-ended text generation tasks, since recently proposed LLMs such as ChatGPT are mainly applied to these tasks.

3. The benchmark datasets in this paper only contain CNN/DM for summarization and MultiWOZ for task-oriented dialogue generation, which are not convincing enough to evaluate the performance of the proposed method based on LLMs. The authors should conduct experiments on broader tasks and datasets. Also, I’m curious about the motivation to choose MultiWOZ. In Section 3.2, the authors say that LLM-based chatbots such as ChatGPT face challenges in handling task-oriented dialogue generation because this task requires the chatbot to respond based on reliable information from API calls or database queries. But the proposed method still cannot interact with APIs or databases. The experimental setting of MultiWOZ seems like end-to-end response generation (i.e. generating the system response given the dialogue history), which is similar to open-domain conversations.

4. From [1], automatic evaluation metrics such as ROUGE cannot reliably evaluate the quality of LLM-generated summaries. Thus, human evaluation should be added to strengthen the experimental results.

[1] News Summarization and Evaluation in the Era of GPT-3.

**Questions:**

I have included my questions in the weaknesses part.

**Limitations:**

The authors should discuss more on the risk of guiding LLMs to generate unethical contents and how to avoid it via the policy model.

---

> ### Author Rebuttal · Authors · 2023-08-07
>
> We would like to express our gratitude for recognizing the strengths in our manuscript and for the detailed feedback. We value these suggestions and hope our response can address the concerns accordingly.
>
> **Response to W1: The technical novelty of the proposed method is limited. The training of the policy model in this paper is a standard paradigm including supervised fine-tuning and reinforcement learning.**
>
> The goal of our work is to tackle the notable challenge of guiding black-box LLM outputs towards desired outcomes for specific tasks, datasets, and even queries. This is a pressing yet challenging issue to address, as it is both inefficient and infeasible to directly fine-tune black-box LLMs. Our proposed framework innovatively proposes to fine-tune a small policy model to guide black-box LLMs, bypassing the constraints of being unable to fine-tune them directly. This can be accomplished with standard SFT/RL training approaches, showcasing the effectiveness and adaptability of our approach. Therefore, introducing new SFT/RL training methods that are not "standard paradigm" is not the paper's focus. It is worth mentioning that the novelty and value have gained recognition from other reviewers.
>
> **Response to W2: general principle to select pseudo-stimulus, especially in open-ended text generation tasks**
>
> A principle is to select latent control codes that can influence the generated outputs. These control codes are intrinsic to open-ended text generation tasks, where models have more freedom to decide the textual content they generate. In our experiments, we choose keywords as they can guide the key points that should be incorporated in the generated summary. Likewise, dialogue acts are used for task-oriented dialogues, which could indicate the appropriate responses given the existing dialogue context. As for other open-ended text generation tasks like open-domain chit-chat, the control codes can be emotions, topics, styles, etc. Similarly, for story generation tasks, potential control codes could include key events and the narrative style, both of which steer the direction and tone of the story. Taking it a step further, large language models (LLMs) can be utilized to automatically analyze and determine the latent control codes specific to certain tasks.
>
> **Response to W3: The benchmark datasets in this paper only contain CNN/DM for summarization and MultiWOZ for task-oriented dialogue generation. Also, I’m curious about the motivation to choose MultiWOZ.**
>
> We have expanded our experiments by incorporating an additional task on the arithmetic reasoning datasets MultiArith [1] and AQuA [2]. The details can be found in the [global author rebuttal](https://openreview.net/forum?id=UvIN8oQ4uI&noteId=ztu9uesRIz).
>
> As for the reason for the choice of MultiWOZ: Currently, most advances in LLMs and chatbots are predominantly in open-domain conversations where they have demonstrated impressive capabilities. However, LLMs still struggle with task-oriented dialogues where they should follow task-specific dialogue flows and output formats, as observed in [1] [2]. Therefore, we choose to fill the gap and experiment on the widely-used task-oriented dialogue dataset MultiWOZ. In addition, the MultiWOZ dataset provides annotations of dialogue acts, which could be directly used as the stimulus in our DSP framework, avoiding the additional annotation efforts.
>
> **Response to W4: human evaluation should be added to strengthen the experimental results.**
>
> We appreciate and value the suggestion. In response, we incorporate evaluations based on GPT-4, which is found to be able to provide consistently high-quality assessments of text generation, being a good alternative to human evaluations. Specifically, we leveraged GPT-4 to compare the summaries generated with our proposed DSP and the original standard prompting based on the assessment of overlap of key points between generated and reference summaries. GPT-4 was instructed to first generate an explanation, followed by the corresponding answer (who wins). The prompt used for the GPT-4 evaluation can be found [here](https://openreview.net/forum?id=UvIN8oQ4uI&noteId=6sYAdkCX9o).
>
> From the 500 test samples: DSP-generated summaries were favored 255 times (**51.0%**), summaries generated with standard prompting were favored 222 times (**44.4%**), while a tie was observed in 23 cases (**4.6%**). We found that GPT-4 can produce reasonable and detailed explanations of their assessment. We will release the GPT-4 evaluation results, including the explanations.
>
> **References**
>
> [1] Hudeček, Vojtěch, and Ondřej Dušek. "Are LLMs All You Need for Task-Oriented Dialogue?." arXiv preprint arXiv:2304.06556 (2023).
>
> [2] Bang, Yejin, et al. "A multitask, multilingual, multimodal evaluation of chatgpt on reasoning, hallucination, and interactivity." arXiv preprint arXiv:2302.04023 (2023).

---

> > ### Comment · Reviewer_2Z7n · 2023-08-21
> > **Response to Rebuttal**
> >
> > Thanks for your rebuttal. My main concern is still about the technical novelty. As I mention in my original review, the novelty of the method is mainly on the applicational side, which means that applying a general method (i.e., SFT+RL) to deal with a new scenario (i.e., guiding black-box LLM outputs towards desired outcomes for specific tasks) is the main focus. I think the applicational papers can also have novel contributions if the authors propose techniques on how to apply the method to the specific task considering the task's characteristics. However, the techniques in this paper are nearly borrowed from existing works, which provide few insights on solving this task.

---

> > > ### Author Response · Authors · 2023-08-21
> > > **Response to the reviewer's reply**
> > >
> > > Thank you for your reply.
> > >
> > > Regarding your concern about technical novelty, we would like to emphasize that our main focus is on addressing the significant and urgent challenge of aligning black-box large language models (LLMs) to specific tasks. Importantly, direct fine-tuning of these LLMs is infeasible and inefficient for most users and researchers, making the issue pressing yet challenging. In light of this, our approach innovatively sidesteps this constraint by proposing the fine-tuning of a smaller policy model, which then guides the black-box LLM, instead of directly fine-tuning them. Given that standard paradigms for tuning these policy models already fulfill our intended objective, there seems little incentive and motivation to introduce new training algorithms for our purpose. Moreover, devising novel training techniques for tuning the black-box LLMs themselves would not address the primary constraint: their inaccessibility to most users and researchers.
> > >
> > > Regarding the application side of our work, we respectfully believe that we have applied our framework to different tasks considering their unique characteristics:
> > > 1. Summarization: By incorporating query-specific keywords, we cue the LLM about essential keypoints that should include, aligning the generated summary more closely with the desired summary.
> > > 2. Dialogue Response Generation: Through the generated dialogue acts, we guide the LLM to generate responses that adhere to desired dialogue/business flows and output formats.
> > > 3. Reasoning: By employing our policy model to generate query-specific prompts, we trigger the LLM to perform chain-of-thought reasoning, mitigating the inconsistencies from manually selecting prompts for different datasets and enhancing reasoning performance.
> > >
> > > We validated the effectiveness of our proposed framework by experimenting on these tasks considering the task characteristics and we believe they are not mirroring previous methods. However, if there are specific works the reviewer feels we are borrowing from, we would greatly appreciate it if the reviewer could point us to them. This would allow us to provide further clarifications or address any oversights.
> > >
> > > We are open to further discussions and remain committed to addressing your concerns.

---

### Official Review · Reviewer_FEWZ · 2023-07-04

**Soundness:** 3 good
**Presentation:** 4 excellent
**Contribution:** 3 good
**Rating:** 6
**Confidence:** 4

**Summary:**

This paper introduces a novel approach to effectively guide black-box LLMs towards generating desired outputs. The proposed approach involves utilizing a relatively small policy model to generate "directional stimulus," which serves as specific information to assist LLMs in performing tasks such as summarization and task-oriented dialogue. The policy model is trained using rewards obtained from evaluating LLM outputs. Notably, this approach requires only a small amount of training data and eliminates the need to train the entire model.

**Strengths:**

- The writing is clear, making it easy to grasp the motivation and contribution of the paper.
- The proposed approach is simple, which suggests its potential for widespread adoption across various domains.
- The experimental results effectively demonstrate the effectiveness of the approach. Notably, in Figure 3, the model leveraging DSP achieves superior performance in terms of BLEU, METEOR, and BERTScore, despite Rouge being used as the reward metric.
- The fact that the authors have released their code is highly valuable, as it facilitates future research and allows for the replication of their findings.

**Weaknesses:**

- I have a concern regarding the generalizability of the method, as the experiments primarily rely on a single prompt per task.
  - It is widely acknowledged that LLMs heavily rely on the structure and format of the prompt, which raises the question of whether this technique would be effective with different prompt formats.
  - This approach would hold even greater value if it could consistently yield improvements regardless of the choice of prompt format, demonstrating its robustness.
- Additional analysis about generated clues will be helpful to get lessons from this work. For example,
  - How does the result change if the number of clues is increased/reduced?
  - What kind of clues are generated considering the semantics/contents of the input (visualization will be helpful)
- The method is not intuitive to me, because all the information that is required to perform the task is already contained in the input. Does the result mean that LLMs are not sufficient to extract information from naive and complex inputs? It will be helpful if the authors provide thoughts about this phenomenon.

**Questions:**

- It would be valuable to know the exact API cost associated with training the reward model. The authors seem to have made efforts to minimize API calls, and providing this information would offer additional details for other researchers to reproduce the study accurately.
- In relation to the aforementioned weakness, it would be interesting to investigate how the model's performance is affected by changing the prompt template. Understanding the impact of prompt variations would provide insights into the robustness of the approach.
- Regarding Figure 1, it appears that the part describing what is "highlighted in blue" is missing. Including this information would clarify the intended meaning.
- As a suggestion, exploring the possibility of allowing the model to "edit" the prompt could potentially offer greater flexibility and a wider search space, leading to improved performance. This suggestion aims to enhance the approach by incorporating additional avenues for improvement.

**Limitations:**

The authors' discussion of the limitations in the paper is not sufficient. Addressing the suggested weaknesses and questions in reviews would significantly enhance the insights provided by this paper.

---

> ### Author Rebuttal · Authors · 2023-08-07
>
> We deeply appreciate your recognition of our work's strengths and the constructive feedback and suggestions. We hope our response can address your concerns accordingly.
>
> **Response to W1: robustness to different prompt formats.**
>
> 1. **Robustness of DSP**: Through the experiment detailed in the "Response to Q1" in the [rebuttal to Reviewer JgjJ](https://openreview.net/forum?id=UvIN8oQ4uI&noteId=6sYAdkCX9o), we found that the policy model trained with few(3)-shot prompting can still improve performance when testing with zero-shot prompting. When both training and evaluation are conducted with zero-shot prompting, the performance improvement over standard prompting is also comparable to both using few-shot prompting (the results shown in our current paper). These observations suggest the robustness of our approach to different numbers of examples in the prompt.
>
> 2. **Addressing prompt sensitivity of LLMs**: Sensitivity to prompts is a critical and inherent challenge of LLMs. Our proposed DSP could be potentially used to find the suitable prompt for each sample. To prove that, we conducted experiments on the chain-of-thought (CoT) reasoning, as detailed in the [global author rebuttal](https://openreview.net/forum?id=UvIN8oQ4uI&noteId=ztu9uesRIz). Current prompting methods typically rely on task-specific prompts. In our experiment, we use DSP to generate query-specific trigger prompts for chain-of-thought reasoning. Our experimental results demonstrated that text-davinci-002 is highly sensitive to the used prompts. DSP improved its zero-shot CoT reasoning performance over 14 tested human-designed prompts and a prompt discovered by the APE approach [1].
>
> **Response to W2: Additional analysis about generated clues will be helpful to get lessons from this work.**
> We included the following analysis of the generated clues/keywords.
> - Number of generated keywords: We've outlined changes in the number of generated keywords, hit keywords (those matched in the reference summary), and corresponding ROUGE-1 scores throughout the training process in the table below. As training progresses, the policy model appears to generate keywords with increasing accuracy, which aligns positively with the increasing ROUGE-1 score. However, it's worth noting that even when keywords are generated with high precision if their quantity is too limited, the performance doesn't necessarily improve.
> | Training iters | #Generated keywords | #Hit keywords | Keyword Precision | ROUGE-1 |
> | ---- | ---- | ---- | ---- | ---- |
> | 0 | 7.986 | 2.936 | 0.367 | 39.30 |
> | 1  | 6.806 | 2.85 | 0.345 | 39.24 |
> | 3 | 7.516 | 3.174 | 0.422 | 39.61 |
> | 5 | 7.262 | 3.31 | 0.456 | 39.63 |
> | 7 | 6.676 | 3.134 | 0.469 | 39.96 |
> | 9 | 6.186 | 2.998 | 0.485 | 39.91 |
> - Type of generated keywords: We employed the spacy package for Part-of-Speech (POS) and Named Entity Recognition (NER) tagging on the generated keywords. The results are shown in the tables below. For the POS tagging, we observe that nouns (NOUN) and proper nouns (PROPN) are the most frequently generated keywords, which can serve as informative keywords. As for the NER tagging, the most commonly generated keywords include persons (PERSON), geopolitical entities (GPE), dates (DATE), organizations (ORG), and numerals (CARDINAL).
>
> | POS Tagging | Appearances | Frequency |
> | ---- | ---- | ---- |
> | NOUN | 2342 | 36.15% |
> | PROPN | 2327 | 35.92% |
> | ADJ | 516 | 7.97% |
> | NUM | 476 | 7.35% |
> | DET | 261 | 4.03% |
> | VERB | 216 | 3.33% |
> | PRON | 81 | 1.25% |
> | ADP | 76 | 1.17% |
> | ADV | 69 | 1.07% |
> | SYM | 42 | 0.65% |
> | AUX | 21 | 0.32% |
> | PART | 18 | 0.28% |
> | CCONJ | 13 | 0.20% |
> | SCONJ | 12 | 0.19% |
> | X | 5 | 0.08% |
> | INTJ | 3 | 0.05% |
>
> | NER Tagging | Appearances | Frequency |
> | ---- | ---- | ---- |
> | PERSON | 567 | 30.32% |
> | GPE | 284 | 15.19% |
> | DATE | 272 | 14.55% |
> | ORG | 247 | 13.21% |
> | CARDINAL | 245 | 13.10% |
> | NORP | 95 | 5.08% |
> | ORDINAL | 49 | 2.62% |
> | MONEY | 37 | 1.98% |
> | TIME | 26 | 1.39% |
> | QUANTITY | 11 | 0.59% |
> | EVENT | 10 | 0.53% |
> | LOC | 9 | 0.48% |
> | FAC | 6 | 0.32% |
> | PRODUCT | 5 | 0.27% |
> | PERCENT | 3 | 0.16% |
> | LANGUAGE | 2 | 0.11% |
> | WORK_OF_ART | 2 | 0.11% |
>
> Visualization (pie chart) of these analyses will be included in the updated version of the paper.
>
> **Response to W3: Does the result mean that LLMs are not sufficient to extract information from naive and complex inputs?**
>
> LLMs have demonstrated their ability to generate high-quality text. However, in open-ended generation tasks, where multiple outputs are valid, LLMs may not consistently produce text that aligns precisely with the desired output for specific tasks and datasets.
> For instance, in the summarization task, LLMs can produce high-quality summaries. Nevertheless, different summarizers might have distinct styles and emphases. Capturing these nuanced variations and specific emphases in different queries becomes challenging when guiding LLMs with a general task-specific prompt.
>
> To address this, our approach trains the policy model on labeled data to learn these subtle signals in the dataset and generate query-specific hints to provide LLMs with fine-grained guidance towards desired outputs for the specific tasks and datasets.
>
> **Response to Q1: the exact API cost**
>
> In our estimation, for experiments on the CNNDM dataset, the training cost per iteration is around \\$1.80, while the evaluation cost on the validation/test set is around \\$2.37. As for the experiments on the MultiWOZ dataset, the training cost is around \\$1.05 per iteration and the evaluation cost is around \\$26.7 for the whole test set.
>
> **Response to Q3: highlighted in blue in Figure 1**
>
> The part highlighted in blue is the generated summary. For DSP, the summary is generated given the provided keywords/hints. For standard prompting, such hints are absent when generating the summary.
>
> **Response to Q4: prompt edit experiments**
>
> Please refer to the global author rebuttal for details on the experiments.

---

### Official Review · Reviewer_RT8N · 2023-07-06

**Soundness:** 3 good
**Presentation:** 3 good
**Contribution:** 3 good
**Rating:** 6
**Confidence:** 4

**Summary:**

In this paper, a prompting framework called Directional Stimulus Prompting (DSP) is proposed which provides a more fine-grained guidance and control over LLMs by adding directional stimulus into the prompt. These directional stimulus or hints are generated by a small tunable model which is fine-tuned using supervised learning and reinforcement learning. The performance of the model is tested on summarization and dialogue response generation tasks.

**Strengths:**

S1: It is interesting that they fine-tuned and used a small language model to improve the performance of larger language models such as ChatGPT. This approach cleverly bypasses the constraint of being unable to fine-tune large language models.

S2: In both experiments, they demonstrated the effectiveness of their approach by showing the improvement achieved through their framework.

S3: The paper is well written and has a good flow.

S4: A good number of related works are covered in the Related work section.



**Weaknesses:**

In their experiments, they only show the results for the flan-T5-large model as a model for generating directional stimulus. They could have also included the performance of other fine-tuned models in their experiments.

**Questions:**

See weakness section.

**Limitations:**

One of the good points that they motioned is that their framework could be used to guide LLMs to generate harmful or biased contents.

---

> ### Author Rebuttal · Authors · 2023-08-07
>
> We genuinely appreciate your acknowledgment of our work's strengths and your valuable feedback.
>
> Regarding the concern about the exclusive presentation of results for the flan-t5-large model, we would like to underscore that our proposed DSP is a general framework and it is not tailored or confined to a specific task or policy model/LLM. The policy model's role in our evaluated tasks is to generate an output given the query context as input. Hence, we chose the suitable widely-used seq2seq t5 models. Due to the enhanced capabilities of flan-t5 models relative to same-sized models, we prioritized it in our experiments.
>
> Additionally, we did try t5-base and flant-t5-base in our preliminary tests and the results consistently demonstrated the advantage of using DSP over standard prompting. For instance, in the MultiWOZ dataset, results using t5-base and flan-t5-base are comparable to those of the flan-t5-large model. With the flan-t5-base and Codex as the LLM, after training on 80 dialogues from MultiWOZ 2.0, we can also achieve a BLEU score of 10.32, success rate of 78.33, inform rate of 91.67, and combined score of 95.32.
> In our new task detailed in the [global author rebuttal](https://openreview.net/forum?id=UvIN8oQ4uI&noteId=ztu9uesRIz), we employed the t5-base as the policy model, and the results also demonstrated the effectiveness of our framework.

---

### Official Review · Reviewer_JgjJ · 2023-07-07

**Soundness:** 3 good
**Presentation:** 3 good
**Contribution:** 3 good
**Rating:** 7
**Confidence:** 3

**Summary:**

This paper introduces a interesting module named 'Directional Stimulus Prompting' (DSP). This innovative module functions by generating cues or hints to aid black-box Large Language Models (LLMs) in response generation. For instance, in a summarization task, providing keywords can guide the LLM towards generating more accurate and relevant answers.

The uniqueness of this module lies in its ability to be fine-tuned through both supervised learning and reinforcement learning techniques. In reinforcement learning, automatic metrics (reward) are applied to responses generated from black-box LLMs, which have been influenced by the DSP.

The results from experiments confirm that both the supervised fine-tuning and reinforcement learning approaches to DSP can significantly enhance the performance of black-box LLMs. This finding underscores the potential of DSP as a powerful tool for improving LLM response generation.

**Strengths:**

This paper introduces a compact yet powerful module known as 'Directional Stimulus Prompting' (DSP). Its function is to generate hints that enhance queries posed to black-box Large Language Models (LLMs).

Two distinct approaches for fine-tuning this DSP module are presented by the authors: supervised fine-tuning and reinforcement learning.

The effectiveness of both supervised fine-tuning DSP (SFT DSP) and reinforcement learning DSP (RL DSP) is confirmed through experimental results, underscoring the utility of this novel module in enhancing LLM query performance.

**Weaknesses:**

Currently, automatic metrics are employed for evaluation, as well as for the reinforcement learning (RL) tuning of the Directional Stimulus Prompting (DSP) module. However, presenting these metrics exclusively in the final analysis could restrict the comprehensive assessment of performance. It may be more insightful to incorporate other evaluation metrics into the analysis. This could include human evaluation, entity word extraction, and GPT-based evaluations. Such a diversified metrics approach could potentially provide a more holistic view of the performance improvements achieved through the use of the DSP module.

**Questions:**

Could the "few-shot prompting" or "in-context learning" potentially enhance the effectiveness of the Directional Stimulus Prompting (DSP) module?

---

> ### Author Rebuttal · Authors · 2023-08-07
>
> We greatly appreciate your feedback and insightful suggestions. Below is our response to address these points.
>
> **Response to W1: exclusive use of automatic metrics in the final analysis**
>
> To address the concern about the exclusive use of automatic metrics in the final analysis, we incorporated GPT-4-based evaluation on the CNNDM summarization dataset. As we employ ROUGE scores as rewards for tuning the policy model to generate keywords that guide the LLM towards generating summaries more aligned with the reference summary, we leveraged GPT-4 to assess the overlap of key points between generated and reference summaries. Specifically, we use GPT-4 to compare the summaries generated with our proposed DSP and the orginal standard prompting. GPT-4 was instructed to first generate an explanation, followed by the corresponding answer. The prompt for the GPT-4 evaluation is as follows:
> ```
> You are provided with an article and a corresponding reference summary. Additionally, there will be two alternative summaries labeled as 'A' and 'B'.
> Your task is to identify which of the two summaries (A or B) is more similar to the reference summary. This similarity should be evaluated based on the presence and accuracy of key points from the reference summary in each alternative summary.
> Please detail your reasoning in an explanation. After your explanation, classify the task outcome as: select 'A wins' if Summary A aligns more closely with the reference summary, 'B wins' if Summary B aligns more closely, or 'Tie' if both summaries align equally well with the reference summary.
> ```
> We found that GPT-4 can produce reasonable and detailed explanations of their assessment. From our test set of 500 samples: DSP-generated summaries were favored 255 times (**51.0%**), summaries generated with original standard prompting were favored 222 times (**44.4%**), while a tie was observed in 23 cases (**4.6%**). We will release the GPT-4 evaluation results, including explanations.
>
> **Response to Q1: Could the "few-shot prompting" or "in-context learning" potentially enhance the effectiveness of the Directional Stimulus Prompting (DSP) module?**
>
> In our current experiments, we employ few-shot prompting with 3 examples in the prompt during training and evaluation. The specific prompt and demonstration examples utilized are detailed in the Appendix. In response to your question, we evaluated two experimental settings on the CNNDM dataset (4,000 training samples):
> 1. Few(3)-shot during training and zero-shot during testing.
> 2. Zero-shot during both training and testing.
>
> The zero-shot evaluation results are as follows:
> | Method | ROUGE-1 | ROUGE-2 | ROUGE-L | BLEU | METEOR | BERTScore |
> | --- | --- | --- | --- | --- | --- | --- |
> | Standard Prompting | 37.34 | 14.13 | 24.43 | 4.86 | 29.85 | 0.8798 |
> | DSP w/ SFT+RL(0-shot) | 38.73 | 15.14 | 25.26 | 5.22 | 31.21 | 0.8820 |
> | DSP w/ SFT+RL(3-shot) | 38.47 | 15.20 | 24.82 | 5.17 | 31.29 | 0.8815 |
>
> From the results, our approach exhibits robustness when different numbers of examples are used in prompts during training and evaluation. When both training and testing are conducted using zero-shot prompting, the performance improvement over standard prompting is also comparable to the scenario where both are conducted using few-shot prompting (results shown in the current paper).

---

> > ### Comment · Reviewer_JgjJ · 2023-08-21
> > **Response to Authors' Rebuttal**
> >
> > Thank you to the authors for their rebuttal. The explanations provided have addressed some of the weaknesses I highlighted. As a result, I have revised my score to 7 (Accept).

---

### Official Review · Reviewer_14f8 · 2023-07-07

**Soundness:** 3 good
**Presentation:** 3 good
**Contribution:** 3 good
**Rating:** 6
**Confidence:** 2

**Summary:**

This paper introduces Directional Stimulus Prompting (DSP), a new prompting framework that introduces directional stimulus into the prompt, which could provide black-box LLMs with fine-grained and query-specific guidance toward the desired outputs.

The experiments on summarization and dialogue response generation tasks demonstrate the effectiveness of this approach. Notably, on the MultWOZ dataset, our framework enables ChatGPT to achieve a remarkable 41.4% improvement in its combined score with only 80 dialogues.

**Strengths:**

1.It is quite novel that the paper proposed DSP, a prompting framework for guiding black-box LLMs toward desired outputs, which combined Supervised fine-tuning and Reinforcement learning to further optimize model.

2.The experiment setting is quite detailed, since this paper used varying numbers of training samples from the datasets and evaluation metrics for ease of display and comparison.

**Weaknesses:**

1.There is too little dataset in the experiment section. Both tasks were conducted on a single dataset and cannot fully demonstrate the effectiveness of the framework.

2.On Page 5 Line 151, the experiment section of the paper is not enough to prove this conclusion. It is not rigorous to say that the framework can be flexibly applied to various types of LMs and generation tasks, just by conducting experiments on two tasks.

**Questions:**

None

---

> ### Author Rebuttal · Authors · 2023-08-07
>
> We are grateful to the reviewer for acknowledging the strengths of our paper and offering valuable feedback. We aim to address your concerns in the following response.
>
> Regarding the concerns about the scope of our experiments, we expanded our experiments with an additional reasoning task on two reasoning datasets MultiArith [1] and AQuA [2] (Details are provided in the [global author rebuttal](https://openreview.net/forum?id=UvIN8oQ4uI&noteId=ztu9uesRIz)).
>
> While current prompting methods typically employ general task-specific prompts, LLMs exhibit sensitivity to these prompts. Consequently, a general prompt might not always be the optimal choice for every scenario.  In our extended experiment, we leverage DSP to generate query-specific trigger prompts for chain-of-thought reasoning. Our results showed that text-davinci-002's zero-shot chain-of-thought reasoning performance is indeed highly sensitive to the used trigger prompts. With the query-specific prompts generated by the policy model, DSP improved text-davinci-002's performance compared with using 14 different human-designed task-specific prompts [3] and also a prompt automatically discovered by the APE approach [4]. This study offers further evidence that our framework can provide LLMs with fine-grained query-specific guidance to align them with desired outputs (i.e., conducting chain-of-thought reasoning to derive the correct answer in this case).
>
> We hope that our response and the expanded experiments can address your concerns.
>
> **References**
>
> [1] Roy, S. and Roth, D. Solving general arithmetic word problems. arXiv preprint arXiv:1608.01413, 2016.
>
> [2] Ling, Wang, et al. "Program induction by rationale generation: Learning to solve and explain algebraic word problems." arXiv preprint arXiv:1705.04146 (2017).
>
> [3] Kojima, T., Gu, S. S., Reid, M., Matsuo, Y., and Iwasawa, Y. Large language models are zero-shot reasoners. Advances in neural information processing systems, 35:22199–22213, 2022.
>
> [4] Zhou, Y., Muresanu, A. I., Han, Z., Paster, K., Pitis, S., Chan, H., and Ba, J. Large language models are human-level prompt engineers. arXiv preprint arXiv:2211.01910, 2022.

---

### Author Rebuttal · Authors · 2023-08-07

### Additional experiment
We greatly appreciate all the reviewers' insightful suggestions and feedback. We conducted an additional experiment in which we use DSP to provide query-specific trigger prompts for chain-of-thought reasoning using two widely-used datasets MultiArith [1] and AQuA [2]. Our results showed that our method DSP could improve text-davinci-002's zero-shot chain-of-thought reasoning performance with query-specific prompts, compared with task-specific human-designed prompts and the prompt automatically discovered with the APE approach [4].

**Experimental setup**: We adopted the experimental setup from previous work [3][4]. We tested the zero-shot chain-of-thought reasoning abilities of text-davinci-002 with different trigger prompt templates. There are 600 examples in the MultiArith dataset, which we divided into 300/50/250 for training/validation/test set. As for the AQuA dataset, we use the standard test set with 254 samples, 300 samples from the standard training set for our training, and 100 samples for the standard validation set for our validation.

**Supervised fine-tuning details**: For supervised fine-tuning (SFT), we first run inference on the training set with the 14 human-designed prompts tested in [3], respectively. We then selected those prompt and query pairs which resulted in a correct chain-of-thought reasoning outcome to form the training set for SFT. These query-prompt pairs were used to train a t5-base policy model for 2 epochs, with the model input being the query instance and the target output a trigger prompt.

**RL training details**: After SFT, the prompts generated by the policy model were used to trigger text-davinci-002 for zero-shot CoT prompting. Reasoning accuracy was utilized as the reward for reinforcement learning (RL). A reward of 1 was assigned for correct reasoning results and 0 otherwise. We conducted 20 training iterations (106k episodes), with 5 epochs per batch, a batch size of 8, and a learning rate of 2e-6. The parameters for $KL_{target}$ and $\beta_0$ were set to 0.5 and 0.001, respectively.

**Results**: We compare the performance of using our generated query-specific prompts with using the 14 human-designed prompts which we used as the pseudo-stimulus to constitute the training set for SFT and also a prompt discovered by the APE approach [4]. Note that all these 15 prompts are general task-specific and are used for the whole test set. By contrast, our proposed DSP enables us to generate query-specific prompts to trigger the LLM's CoT reasoning. The performance comparison is shown in the table below.

| No. | Category | Zero-shot CoT Trigger Prompt | MultiArith | AQuA |
| ---- | ---- | ---- | ---- | ---- |
| 1 | Human-Designed | Let's think step by step. | 79.6 | 31.9 |
| 2 | Human-Designed | We should think about this step by step. | 81.2 | 28.7 |
| 3 | Human-Designed | First, | 78.0 | 38.2 |
| 4 | Human-Designed | Before we dive into the answer, | 54.8 | 27.2 |
| 5 | Human-Designed | Proof followed by the answer. | 58.4 | 37.8 |
| 6 | Human-Designed | Let's think step by step in a realistic way. | 59.6 | 33.9 |
| 7 | Human-Designed | Let's think step by step using common sense and knowledge. | 80.0 | 34.3 |
| 8 | Human-Designed | Let's think like a detective step by step. | 73.6 | 24.0 |
| 9 | Human-Designed | Let's think about this logically. | 75.2 | 34.7 |
| 10 | Human-Designed | Let's think step by step. First, | 78.8 | 32.3 |
| 11 | Human-Designed | Let's think | 56.8 | 38.2 |
| 12 | Human-Designed | Let's solve this problem by splitting it into steps. | 72.4 | 32.3 |
| 13 | Human-Designed | The answer is after the proof. | 42.8 | 34.3 |
| 14 | Human-Designed | Let's be realistic and think step by step. | 69.6 | 29.9 |
| 15 | APE [4] | Let's work this out in a step by step way to be sure we have the right answer. | 81.6 | 34.3 |
| 16 | DSP w/ SFT | *Query-specific* | 75.2 | 35.8 |
| 17 | DSP w/ SFT+RL | *Query-specific* | **82.4** | **38.6** |

As can be seen, text-davinci-002's performance varies significantly when using different task-specific prompts. Compared to the 14 task-specific human-designed prompts, DSP enhances the performance of text-davinci-002 with query-specific prompts. It also outperforms the prompt discovered by the APE approach [4]. Solely relying on supervised fine-tuning of the policy model with the dataset comprising the 14 human-designed prompts doesn't lead to its peak performance. After fine-tuning with RL, the policy model is encouraged to explore better query-specific trigger prompts, further improving performance. Some of the newly generated trigger prompts include:
 - "*Let's think like a detective step by step. First,*"
- "*Let's solve this problem by splitting it into steps. First,*"
- "*First step:*"
- "*Let’s think step by step. First*",
- "*Let's think step by step using our creative brains.*"
- "*Let's think step by step using both the above information and the testing.*"
- "*Let's think step by step using proven methods.*"

Overall, the results provide further evidence that DSP could provide fine-grained query-specific guidance to LLMs to align them better with the desired output (chain-of-thought reasoning with correct answers in this case).
Further details will be provided in the updated version of our paper.


**References**:

[1] Roy, Subhro, and Dan Roth. "Solving general arithmetic word problems." arXiv preprint arXiv:1608.01413 (2016).

[2] Ling, Wang, et al. "Program induction by rationale generation: Learning to solve and explain algebraic word problems." arXiv preprint arXiv:1705.04146 (2017).

[3] Kojima, T., Gu, S. S., Reid, M., Matsuo, Y., and Iwasawa, Y. Large language models are zero-shot reasoners. Advances in neural information processing systems, 35:22199–22213, 2022.

[4] Zhou, Y., Muresanu, A. I., Han, Z., Paster, K., Pitis, S., Chan, H., and Ba, J. Large language models are human-level prompt engineers. arXiv preprint arXiv:2211.01910, 2022.

---

### Decision · Program_Chairs · 2023-09-21

**Decision:**

Accept (poster)

**Comment:**

One consistent strength highlighted by multiple reviewers is the 'Directional Stimulus Prompting' (DSP) module. It offers an approach to guiding black-box Large Language Models (LLMs) by generating hints or cues, and the combination of supervised fine-tuning and reinforcement learning techniques adds to its innovative appeal. The experimental setup in the paper has also been appreciated for its detail, using various training sample sizes and evaluation metrics for clarity and comparison. The authors conducted some addition experiments during rebuttal on MultiArith and AQuA to show its good performance.

However, there are shared concerns among the reviewers, in order to make the paper more comprehensive. The limited dataset used in the experiments, with a focus on a single dataset for both summarization and dialogue generation, raises questions about the generalizability of DSP to a broader range of tasks. Additionally, some reviewers express doubts about the robustness of the method, as it heavily relies on the format and structure of the prompt, making its effectiveness potentially dependent on specific prompt formats. Reviewers suggest the need for a more diverse set of evaluation metrics, including human evaluation, entity word extraction, and GPT-based evaluations, to provide a more comprehensive assessment of DSP's performance (In my opinion, especially human preference is very important, because we have been observing in summarization field that even though fitting into some formats or styles of a dataset, the output summaries are still not preferred by human). Furthermore, there's a call for additional analysis of the generated cues and clues, exploring variations in the number and content of cues.